# UnCLe: Towards Scalable Dynamic Causal Discovery in Non-linear Temporal Systems

**Tingzhu Bi,  Yicheng Pan,  Xinrui Jiang,  Huize Sun,  Meng Ma,**[*] **Ping Wang**

Peking University

bitingzhu@stu.pku.edu.cn, {aqpyc,jxrjxrjxr}@pku.edu.cn

sunhuize@stu.pku.edu.cn, {mameng,pwang}@pku.edu.cn

Code & Datasets: https://github.com/etigerstudio/uncle-causal-discovery

## Abstract

Uncovering cause-effect relationships from observational time series is fundamental to understanding complex systems. While many methods infer static causal graphs, real-world systems often exhibit *dynamic causality*—where relationships evolve over time. Accurately capturing these temporal dynamics requires time-resolved causal graphs. We propose UnCLe, a novel deep learning method for scalable dynamic causal discovery. UnCLe employs a pair of Uncoupler and Recoupler networks to disentangle input time series into semantic representations and learns inter-variable dependencies via auto-regressive Dependency Matrices. It estimates dynamic causal influences by analyzing datapoint-wise prediction errors induced by temporal perturbations. Extensive experiments demonstrate that UnCLe not only outperforms state-of-the-art baselines on static causal discovery benchmarks but, more importantly, exhibits a unique capability to accurately capture and represent evolving temporal causality in both synthetic and real-world dynamic systems (e.g., human motion). UnCLe offers a promising approach for revealing the underlying, time-varying mechanisms of complex phenomena.

## 1 Introduction

Understanding the intricate web of cause-effect relationships is fundamental to unraveling the mechanisms of real-world complex systems, from climate patterns and biological processes to economic or network fluctuations and human biomechanics [20, 28]. A critical, yet often overlooked, aspect is that these systems are inherently dynamic, with causal influences frequently evolving over time due to changing internal states or external conditions. For instance, predator-prey dynamics can shift seasonally, gene regulatory networks can alter during developmental stages, and the biomechanical interplay between human joints changes distinctly across different phases of motion. Accurately capturing these dynamic causal structures through time-resolved causal graphs is therefore essential for achieving a deeper, more veridical understanding, enabling more precise predictions and potentially more effective interventions. The practical success of specialized dynamic models in high-stakes domains, such as real-time fault diagnosis in data centers [2], underscores this urgent need. However, the predominant paradigm in temporal causal discovery has largely focused on inferring static causal graphs, which represent an aggregated or time-averaged view of dependencies, thereby obscuring the rich, evolving nature of causality in many real-world phenomena.

While foundational approaches to temporal causal discovery, such as Granger causality [7] and its various linear (e.g., VAR-based) and nonlinear extensions (e.g., constraint-based methods like PCMCI [22], or early neural network adaptations [27, 17]), have laid crucial groundwork for inferring causal links from time series data, they are often not inherently designed to explicitly model or represent how these causal relationships themselves change over time. Some methods might capture

---

[*]Corresponding author: mameng@pku.edu.cn.

39th Conference on Neural Information Processing Systems (NeurIPS 2025).

lagged effects or offer a global summary graph, but the challenge of constructing and interpreting time-resolved causal graphs—where the set of active causal edges can vary from one time point or interval to another—remains a significant hurdle. This limitation hinders our ability to fully comprehend systems where causal laws are not fixed but adapt, shift, or switch, which is characteristic of many complex adaptive systems.

To bridge this critical gap, we propose UnCLe (UnCoupLing causality), a novel deep learning framework specifically engineered for the scalable discovery and representation of dynamic causal graphs from observational time series. UnCLe's core innovation lies in its ability to first disentangle complex, multivariate time series into meaningful semantic channels using a pair of parameter-sharing Uncoupler and Recoupler networks, and then to model evolving inter-variable dependencies within these channels via auto-regressive Dependency Matrices. Crucially, UnCLe infers time-resolved causal influences by meticulously analyzing datapoint-wise prediction errors that are induced by targeted temporal perturbations of individual series. This mechanism allows UnCLe to quantify how the predictive relationship between variables changes at different points in time, thus constructing a dynamic causal narrative. Furthermore, UnCLe is designed with scalability in mind, enabling its application to large-scale, non-linear systems commonly encountered in real-world applications.

The main contributions of this paper are:

- We propose UnCLe, a novel and scalable deep learning method for dynamic temporal causal discovery, capable of generating and representing time-resolved causal graphs that capture evolving cause-effect relationships.

- We introduce a methodology that combines semantic disentanglement of time series with perturbation-based, datapoint-wise error analysis to effectively identify and quantify dynamic causal influences.

- We demonstrate UnCLe's superior ability to uncover and track evolving causal structures through extensive experiments on synthetic datasets with known dynamic ground truths (e.g., time-varying SEMs) and challenging real-world systems, notably human motion capture (MoCap) data, where UnCLe provides interpretable, phase-specific biomechanical insights.

- We show that UnCLe also achieves competitive or state-of-the-art performance on standard static causal discovery benchmarks, highlighting its versatility and robustness.

By offering a principled and effective approach to dynamic causal discovery, UnCLe aims to provide a more powerful lens for understanding the complex, ever-changing mechanisms that govern the world around us.

## 2 Background and Related Work

**Notations** A *dynamic causal graph* is defined as a time-varying graph $\mathcal{G}^t = \{\mathcal{V}^t, \mathcal{E}^t\}$ for each timestep $t \in \{1, \ldots, T\}$, where $\mathcal{V}^t = \{v_i^t \mid v_i^t \in \mathcal{V}\}$ represents the set of vertices at time $t$, corresponding to time series $\boldsymbol{x}_i^t$ observed at time $t$, and $\mathcal{E}^t = \{(v_i^t, v_j^t) \mid v_i^t \in \mathcal{V}^t, v_j^t \in \mathcal{V}^t\}$ represents the set of directed edges at time $t$. An edge $(v_i^t, v_j^t)$ denotes a causal-effect relationship, where $v_i^t$ is the cause variable and $v_j^t$ is the effect variable at time $t$. A *static causal graph* [20] is defined as a time-invariant graph $\mathcal{G} = \{\mathcal{V}, \mathcal{E}\}$ whose variables and cause-effect relationship remain constant over time.

**Granger causality** Granger causality [7] is a widely used statistical framework for defining causality based on predictive relationships. It is grounded in the intuition that a time series $\boldsymbol{x}_i$ can be considered a cause of another time series $\boldsymbol{x}_j$ if the inclusion of $\boldsymbol{x}_i$'s past values improves the prediction of $\boldsymbol{x}_j$'s future values. Formally, the generalized form of Granger causality can be expressed as follows [24]:

$$\boldsymbol{x}_{i,t} = h_i\left(\boldsymbol{x}_{1,<t}, \ldots, \boldsymbol{x}_{N,<t}\right) + \epsilon_{i,t},$$

where $h_i$ is a prediction function that maps the past values of all $N$ time series to the current value of series $\boldsymbol{x}_i$, and $\epsilon_{i,t}$ represents the prediction error.

Traditional Granger causal discovery methods typically employ statistical autoregressive (AR) models for $h_i$ and use statistical significance tests on the prediction error $\epsilon$ to infer causal relationships between

time series. In contrast, recent approaches leverage neural networks to model $h_i$ and determine causal relationships through various mechanisms.

**Neural Granger Causality**  Neural Granger causality methods leverage a variety of neural network architectures as their backbone networks, including MLPs [27, 29], RNNs [10], LSTMs [27], CNNs [17], and GNNs [15, 5], with some works successfully employing TCN-based autoencoders for representation learning in specific causality-driven applications like root cause diagnosis [2]. Additionally, design concepts such as attention mechanisms [17], variational autoencoders [12], self-explaining neural networks [16], and inductive modeling [15] are actively incorporated into these methods. The standard procedure for neural Granger causality analysis involves training prediction models using neural networks and then inferring the causal structure from the learned models through various techniques.

For instance, NeuralGC [27] and GVAR [16] analyze the weights of specific network layers to interpret the influence relationships between variables. In contrast, TCDF [17] identifies causal dependencies using attention scores and quantifies the predictive contribution of variables by computing permutation importance [3]. However, NeuralGC, GVAR, and TCDF face significant scalability challenges on large-scale datasets due to their component-wise design, which lacks parameter sharing. This results in $O(N^2)$ parameters to train as the number of variables increases.

To address the scalability problem, both JRNGC [29] and CUTS+ [5] utilize parameter sharing, making them more suitable for large-scale datasets. JRNGC incorporates an input-output Jacobian regularizer into the training objective to learn Granger causality, while CUTS+ enhances scalability on high-dimensional temporal data by splitting time series into groups and applying a coarse-to-fine filtering strategy.

While some methods, such as NeuralGC, JRNGC, and TCDF, support time-lag recognition, only GVAR is capable of generating dynamic causal graphs. Furthermore, to the best of our knowledge, no existing method has been rigorously evaluated on dynamic causal datasets to assess its ability to identify causal evolutions over time.

## 3   Methodology

We introduce UnCLe, a scalable method for dynamic causal discovery rooted in the principles of neural Granger causality. The overall framework is depicted in Figure 1 and comprises two primary phases:

1. *Training Phase.* Given an input multivariate time series dataset $x \in \mathbb{R}^{N \times T}$ (represented as green blocks in Figure 1, denoting data for $N$ variables over $T$ timesteps), UnCLe first trains its core architecture. This architecture consists of a pair of parameter-sharing Uncoupler and Recoupler networks, along with a set of auto-regressive Dependency Matrices ($\Psi$). The Uncoupler transforms the input series $x$ into multi-channel semantic representations $z \in \mathbb{R}^{N \times T \times C}$ (visualized as stacked, colorful blocks for $C$ semantic channels). This transformation is learned through a *reconstruction task*, where the Recoupler aims to reconstruct the original series $\tilde{x}$ from $z$. Concurrently, the Dependency Matrices $\Psi$ are optimized via a *prediction task* to capture inter-variable dependencies within each semantic channel, forecasting future latent representations $\hat{z}$ which are then mapped back to the original space as $\hat{x}$.

2. *Post-hoc Analysis Phase.* Subsequent to training, UnCLe infers causal relationships. For *dynamic causal discovery*, individual time series $x_i$ (denoting the $i$-th variable's series) within the input dataset are chronologically perturbed (e.g., via permutation, resulting in $x^{\setminus j}$, shown as red blocks) to disrupt their temporal structure and thereby diminish their predictive utility. The resulting datapoint-wise increase in prediction error for other series $x_i$ is then quantified as the strength of the dynamic causal link from $x_j$ to $x_i$, forming the dynamic causal graph $\hat{\mathcal{G}}^{\mathrm{Pert}}$. Furthermore, for *static causal discovery*, the learned weights of the Dependency Matrices $\Psi$ are aggregated (e.g., via average pooling) to derive a summary static causal graph $\hat{\mathcal{G}}^{\mathrm{Agg}}$.

The subsequent subsections provide detailed expositions of the UnCLe model architecture and the causal inference procedures.

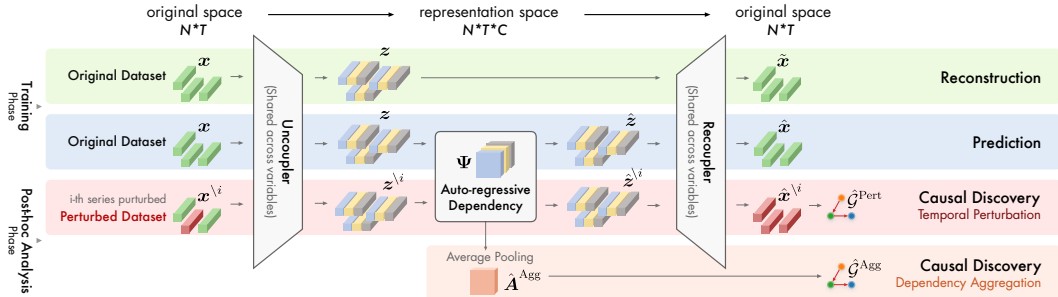

Figure 1: The UnCLe framework. Training involves reconstruction and prediction using Uncoupler/Recoupler and Dependency Matrices ($\mathbf{\Psi}$). Post-hoc analysis uses temporal perturbation for dynamic graphs ($\hat{\mathcal{G}}^{\text{Pert}}$) and aggregation of $\mathbf{\Psi}$ for static graphs ($\hat{\mathcal{G}}^{\text{Agg}}$).

## 3.1 Model Architecture

**Uncoupler and Recoupler Networks**   The Uncoupler and Recoupler form the backbone of UnCLe's representation learning, functioning akin to a parameter-sharing Temporal Convolutional Network (TCN) autoencoder [1]. Their primary role is to model intra-variable temporal dynamics and disentangle the input time series $\mathbf{x} \in \mathbb{R}^{N \times T}$ into meaningful latent representations. By sharing parameters across all $N$ variables, this design significantly enhances learning efficiency, model stability, and the quality of learned representations, especially for high-dimensional data.

The Uncoupler, denoted as $\text{TCN}_{\text{Unc}}(\cdot; \phi_{\text{Unc}})$, maps each univariate time series $\mathbf{x}_i \in \mathbb{R}^T$ (the $i$-th row of $\mathbf{x}$) into a $C$-channel latent sequence $\mathbf{z}_i \in \mathbb{R}^{T \times C}$. Collectively, for all variables, this yields $\mathbf{z} \in \mathbb{R}^{N \times T \times C}$. The Recoupler, $\text{TCN}_{\text{Rec}}(\cdot; \phi_{\text{Rec}})$, then aims to reconstruct the original series $\tilde{\mathbf{x}}_i \in \mathbb{R}^T$ from its corresponding latent sequence $\mathbf{z}_i$. This reconstruction process is formalized as:

$$\mathbf{z}_i = \text{TCN}_{\text{Unc}}(\mathbf{x}_i; \phi_{\text{Unc}}), \quad \tilde{\mathbf{x}}_i = \text{TCN}_{\text{Rec}}(\mathbf{z}_i; \phi_{\text{Rec}}) \tag{1}$$

where $\phi_{\text{Unc}}$ and $\phi_{\text{Rec}}$ represent the learnable parameters of the Uncoupler and Recoupler, respectively. The objective for the reconstruction task is to minimize the Mean Squared Error (MSE) loss:

$$\mathcal{L}_{\text{Recon}}(\phi_{\text{Unc}}, \phi_{\text{Rec}}) = \frac{1}{NT} \sum_{i=1}^{N} \sum_{t=1}^{T} (\tilde{x}_{i,t} - x_{i,t})^2 \tag{2}$$

The TCN architecture, characterized by stacked dilated causal convolution blocks [18], ensures that information processing is strictly temporal (no leakage from future to past), a crucial property for subsequent causal discovery. Furthermore, the parallelizable nature of TCN computations contributes to UnCLe's efficiency on large-scale datasets.

**Auto-regressive Dependency Matrices**   To model inter-variable dependencies, UnCLe introduces a set of $C$ lightweight Dependency Matrices, $\mathbf{\Psi} = \{\mathbf{\Psi}^1, \ldots, \mathbf{\Psi}^C\}$, where each $\mathbf{\Psi}^c \in \mathbb{R}^{N \times N}$. These matrices operate on the disentangled latent representations $\mathbf{z}$ to perform auto-regressive prediction. Specifically, for each semantic channel $c$, the latent representation at the next timestep, $\hat{\mathbf{z}}^c_{:,t+1} \in \mathbb{R}^N$, is predicted from the current latent representations across all variables in that channel, $\mathbf{z}^c_{:,t} \in \mathbb{R}^N$:

$$\hat{\mathbf{z}}^c_{:,t+1} = \sigma(\mathbf{\Psi}^c \mathbf{z}^c_{:,t}) \tag{3}$$

where $\mathbf{z}^c_{:,t}$ denotes the $N$-dimensional vector of latent features for channel $c$ at time $t$, and $\sigma$ denotes the same activation function as TCNs. This linear update is motivated by the principle that a suitable coordinate transformation, learned here by our Uncoupler, can approximate complex non-linear dynamics with a linear system [26, 4].

The predicted latent sequences $\hat{\mathbf{z}} = \{\hat{\mathbf{z}}^1, \ldots, \hat{\mathbf{z}}^C\}$ are then fed into the (shared) Recoupler network to generate predictions in the original data space:

$$\hat{\mathbf{x}}_{:,t+1} = \text{TCN}_{\text{Rec}}(\{\hat{\mathbf{z}}^1_{:,t+1}, \ldots, \hat{\mathbf{z}}^C_{:,t+1}\}; \phi_{\text{Rec}}) \tag{4}$$

The prediction loss $\mathcal{L}_{\text{Pred}}$ is also an MSE loss, calculated between the predicted values $\hat{x}_{i,t+1}$ and the true future values $x_{i,t+1}$:

$$\mathcal{L}_{\text{Pred}}(\phi_{\text{Unc}}, \phi_{\text{Rec}}, \mathbf{\Psi}) = \frac{1}{N(T-1)} \sum_{i=1}^{N} \sum_{t=1}^{T-1} (\hat{x}_{i,t+1} - x_{i,t+1})^2 \tag{5}$$

**Regularization** To mitigate overfitting and discourage the discovery of overly complex causal structures or spurious inter-variable relationships, UnCLe incorporates several regularization techniques. First, L1 regularization is applied to the Dependency Matrices $\boldsymbol{\Psi}$ to promote sparsity in the learned inter-variable connections:

$$\mathcal{L}_{\text{L1}}(\boldsymbol{\Psi}) = \lambda_1 \sum_{c=1}^{C} \sum_{k=1}^{N} \sum_{l=1}^{N} |\Psi_{k,l}^c| \tag{6}$$

where $\lambda_1$ is the L1 regularization hyperparameter. Additionally, to enhance the robustness of the feature disentanglement process within the TCNs, dropout with a rate of 0.2 is applied during the training of the Uncoupler and Recoupler networks.

**Overall Training Objective and Procedure** UnCLe is trained in two stages. In the *pretraining stage*, the model focuses on representation learning by optimizing only the reconstruction loss $\mathcal{L}_{\text{Recon}}$. This stage trains $\phi_{\text{Unc}}$ and $\phi_{\text{Rec}}$, providing a strong initialization for the subsequent phase. In the *full model training stage*, all components, including the Dependency Matrices $\boldsymbol{\Psi}$, are trained jointly by minimizing a composite loss function:

$$\mathcal{L}_{\text{Total}} = \mathcal{L}_{\text{Recon}} + \alpha \mathcal{L}_{\text{Pred}} + \mathcal{L}_{\text{L1}} \tag{7}$$

where $\alpha$ is a hyperparameter balancing the prediction task's contribution. This joint optimization allows the model to simultaneously learn to represent the data, predict its future, and identify underlying dependencies.

## 3.2 Post-hoc Causal Discovery

Once the UnCLe model is trained, causal relationships are inferred in a post-hoc analysis phase using two distinct approaches.

**Perturbation-based Dynamic Granger Causality** To uncover dynamic causal influences, UnCLe employs a temporal perturbation strategy. The core idea is that because the trained model has learned an approximation of the data's causal generative mechanism, disrupting the temporal structure of a true cause $\boldsymbol{x}_j$ will violate the learned dynamics and significantly impair the model's ability to predict its effect $\boldsymbol{x}_i$. Formally, let $\boldsymbol{x} \in \mathbb{R}^{N \times T}$ be the original multivariate time series dataset. We denote by $\boldsymbol{x}^{\backslash j}$ the dataset where the $j$-th time series, $\boldsymbol{x}_j$, has been perturbed by a random permutation of its temporal values. This permutation preserves the marginal distribution and statistical properties of $\boldsymbol{x}_j$ but destroys its original sequential order, thus nullifying its valid predictive information for other series that depend on its specific temporal evolution.

Let $f(\cdot)$ represent the trained UnCLe prediction model (Equations 3-4). The prediction for $\boldsymbol{x}_{i,t}$ using the original dataset is $\hat{x}_{i,t}$. The original prediction error for $\boldsymbol{x}_{i,t}$ can be defined, for instance, as the squared error:

$$\epsilon_{i,t} = (\hat{x}_{i,t} - x_{i,t})^2 \tag{8}$$

When $\boldsymbol{x}_j$ is perturbed to create $\boldsymbol{x}^{\backslash j}$, the model yields a new prediction $\hat{x}_{i,t}^{\backslash j}$ for $\boldsymbol{x}_{i,t}$. The prediction error under this perturbation is:

$$\epsilon_{i,t}^{\backslash j} = (\hat{x}_{i,t}^{\backslash j} - x_{i,t})^2 \tag{9}$$

The increase in prediction error, or error gain, quantifies the causal influence of $\boldsymbol{x}_j$ on $\boldsymbol{x}_i$ at time $t$:

$$\Delta \epsilon_{i,t}^{\backslash j} = \max(0, \epsilon_{i,t}^{\backslash j} - \epsilon_{i,t}) \tag{10}$$

This value, $\Delta \epsilon_{i,t}^{\backslash j}$, represents the strength of the causal link from $\boldsymbol{x}_j$ to $\boldsymbol{x}_i$ specifically at time $t$, forming an element of the time-resolved adjacency matrix $\hat{\boldsymbol{A}}_{j,i}^{t,\text{Pert}}$ of the dynamic causal graph $\hat{\mathcal{G}}^{\text{Pert}}$. UnCLe computes these pairwise error gains for all variable pairs and timesteps. Since the errors are computed at each timestep, this perturbation-based approach inherently captures dynamic causality, allowing causal relationships to evolve over time. For a static summary, these dynamic strengths can be aggregated across the time axis (e.g., by averaging or summing $\Delta \epsilon_{i,t}^{\backslash j}$ over $t$). The detailed algorithm for dynamic causal discovery via temporal perturbation is listed as Algorithm 1 in Appendix L. The batch processing of these computations significantly enhances the efficiency of the causal discovery process.

**Static Causal Graph via Dependency Aggregation** A static, or summary, causal graph $\hat{\mathcal{G}}^{\text{Agg}}$ can also be directly inferred from the learned Dependency Matrices $\Psi$. The rationale is that if variable $x_k$ does not influence $x_l$ in channel $c$, the L1 regularization (Equation 6) will drive the corresponding coefficient $\Psi^c_{l,k}$ towards zero. Conversely, a significant non-zero coefficient suggests a dependency. The elements of the multi-channel Dependency Matrices $\Psi$ are thus interpreted as causal strengths. The aggregated static causal influence from $x_k$ to $x_l$, denoted by $\hat{A}^{\text{Agg}}_{l,k}$, is obtained by pooling the magnitudes of these coefficients across all $C$ channels. A common pooling method is the L2-norm (root mean square) of the coefficients:

$$\hat{A}^{\text{Agg}}_{l,k} = \sqrt{\frac{1}{C}\sum_{c=1}^{C}(\Psi^c_{l,k})^2} \tag{11}$$

This aggregation yields a single $N \times N$ adjacency matrix representing the overall static causal structure.

UnCLe thus offers two complementary modes for causal discovery: (P) Temporal Perturbation, which yields dynamic causal graphs and is generally more accurate as it leverages the full model including the learned representations from the Uncoupler/Recoupler and the input data characteristics. (A) Dependency Aggregation, which produces a static causal graph more rapidly as it directly uses the learned $\Psi$ without further post-hoc predictive analysis.

## 4 Experiments

### 4.1 Experimental Setup

**Datasets** We evaluate UnCLe using various synthetic / real-world datasets from a great variety domains. Apart from other existing datasets, we propose NC8 (Non-linear, Constant connections, 8 variables) and ND8 (Non-linear, Dynamic connections, 8 variables) to better evaluate causal discovery methods. The detailed dataset setup is included in Appendix A.

**Baselines** We compare UnCLe against nine baseline methods spanning a range of categories, including constraint-based, score-based, and cutting-edge neural Granger approaches: (i) VAR, the classic Granger causality method [7] based on pairwise VAR F-tests; (ii) PCMCI, a constraint-based approach [22] that uses partial correlation for independence tests; (iii) cMLP [27], a neural Granger causality method that relies on MLP prediction networks and sparse-inducing regularization; (iv) TCDF [17], which interprets attention weights and validates them using permutation importance; (v) GVAR [16], which leverages neural network-generated dynamic VAR coefficients; (vi) VAR-LiNGAM [9], a method that uses a non-Gaussian structural vector autoregressive model to assess the significance of causal influences; (vii) DYNOTEARS [19], a score-based method that minimizes a penalized loss subject to an acyclicity constraint; and (viii) CUTS+ [5], a neural Granger causality method that utilizes passing-based graph neural networks and supports high-dimensional data; and (ix) JRNGC [30], which employs an input-output Jacobian regularizer to learn causality from a single, shared prediction model. Note that GVAR is the only baseline method that can generate dynamic causal graphs.

### 4.2 Results

We first report the causal discovery accuracy on static graphs. Next, we evaluate dynamic causal discovery performance on two datasets. Finally, we present results on two large-scale real-world transportation datasets. The best accuracies are bolden and the second best are underlined, and "-" indicates the running time of the method exceeded the reasonable time limit.

**Static: Lorenz 96** Table 1 reports the causal discovery performance of UnCLe and other baseline methods on synthetic datasets. *Lorenz96* [14] is a ODE model to simulate climate dynamics used by [16, 27, 10, 29, 5]. The system dynamics increasingly chaotic and thus hard to model for higher values of forcing constant $F$. We design three sets of system configurations of Lorenz96, setting number of variables $p = \{20, 20, 100\}$, timesteps $T = \{250, 250, 500\}$, force $F = \{10, 10, 40\}$ for Lorenz#1, #2, and #3 respectively. On all Lorenz datasets, UnCLe(P) perturbation consistently achieves the highest AUROC and AUPRC scores. The added chaotic strength of Lorenz#2 and large number of variables of Lorenz#3 pose significant challenges to baseline methods. Notably, DYNOTEARS struggles with large-scale datasets, and its DAG constraint conflicts with the ground truth causal structure of the Lorenz system.

Table 1: Static causal discovery performance comparison on synthetic datasets.

| Methods | Lorenz#1 | | Lorenz#2 | | Lorenz#3 | | NC8 | | FINANCE | |
|---|---|---|---|---|---|---|---|---|---|---|
| | AUROC ↑ | AUPRC ↑ | AUROC ↑ | AUPRC ↑ | AUROC ↑ | AUPRC ↑ | AUROC ↑ | AUPRC ↑ | AUROC ↑ | AUPRC ↑ |
| VAR | .853(±.007) | .485(±.012) | .709(±.015) | .295(±.025) | .798(±.017) | .142(±.014) | .633(±.202) | .218(±.200) | .630(±.138) | .103(±.140) |
| PCMCI | .833(±.025) | .482(±.066) | .670(±.027) | .269(±.038) | .712(±.008) | .084(±.009) | .895(±.052) | .408(±.168) | .589(±.154) | .055(±.048) |
| cMLP | .994(±.004) | .975(±.013) | .885(±.016) | .370(±.054) | .814(±.019) | .465(±.041) | .928(±.030) | .717(±.081) | .619(±.108) | .069(±.052) |
| GVAR | .974(±.014) | .878(±.071) | .839(±.027) | .613(±.075) | .558(±.015) | .040(±.003) | .956(±.024) | .831(±.044) | .999(±.001) | .990(±.020) |
| TCDF | .727(±.047) | .345(±.043) | .615(±.019) | .225(±.030) | .885(±.033) | .370(±.109) | .620(±.058) | .279(±.124) | .915(±.003) | .509(±.050) |
| VARLiNGAM | .854(±.066) | .721(±.130) | .627(±.029) | .481(±.040) | .673(±.018) | .383(±.043) | .880(±.060) | .586(±.036) | .946(±.040) | .337(±.149) |
| DYNOTEARS | .544(±.055) | .315(±.049) | .546(±.018) | .240(±.022) | - | - | .546(±.018) | .240(±.022) | - | - |
| CUTS+ | .947(±.033) | .800(±.090) | .894(±.034) | .620(±.056) | .863(±.030) | .247(±.052) | .777(±.032) | .297(±.119) | .885(±.033) | .370(±.109) |
| JRNGC | .983(±.002) | .714(±.024) | .807(±.015) | .266(±.044) | .612(±.014) | .018(±.001) | .756(±.010) | .162(±.006) | .688(±.410) | .714(±.294) |
| UnCLe(P) | .999(±.002) | .996(±.008) | .940(±.011) | .804(±.036) | .922(±.012) | .636(±.071) | .975(±.004) | .835(±.056) | .987(±.041) | .933(±.141) |
| UnCLe(A) | .994(±.007) | .962(±.054) | .871(±.023) | .531(±.080) | .865(±.024) | .356(±.053) | .952(±.035) | .770(±.178) | .972(±.087) | .887(±.283) |

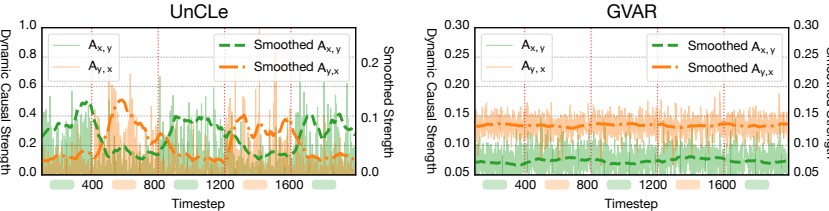

Figure 2: The dynamic causal strengths between $X_t$ and $Y_t$ discovered by UnCLe and GVAR.

**Static: NC8 and FINANCE**  The NC8 dataset evaluates the ability of methods to uncover long-term nonlinear relationships between variables, and UnCLe delivers the best AUROC and AUPRC scores. On the FINANCE dataset, UnCLe demonstrates the second-best performance.

**Dynamic: Time-variant SEM**  To evaluate the dynamic causal discovery capability of UnCLe, we construct a bivariate time-varying Structural Equation Model (TVSEM) with a total length of $T = 2000$ time points. The model is defined as:

$$a_t = \begin{cases} 0.8 & \text{if } \lfloor (t-1)/400 \rfloor \pmod 2 = 0 \\ 0.2 & \text{if } \lfloor (t-1)/400 \rfloor \pmod 2 = 1 \end{cases}, \ b_t = \begin{cases} 0.1 & \text{if } \lfloor (t-1)/400 \rfloor \pmod 2 = 0 \\ 0.7 & \text{if } \lfloor (t-1)/400 \rfloor \pmod 2 = 1 \end{cases} \quad (12)$$

$$X_t = a_t Y_{t-1} + \epsilon_{X,t}, \quad Y_t = b_t X_{t-1} + \epsilon_{Y,t} \quad (13)$$

Here, $X_t$ and $Y_t$ represent the observed variables at time $t$. The error terms $\epsilon_{X,t}$ and $\epsilon_{Y,t} \sim N(0, 0.1)$. The model's coefficients $a_t$ and $b_t$ switch values every 400 time points, creating five segments. This switching pattern is governed by the parity of the segment index $\lfloor (t-1)/400 \rfloor$. When the index is even (segments 1, 3, 5), $(a_t, b_t) = (0.8, 0.1)$, indicating a dominant $Y \to X$ causal direction due to the strong influence from $Y_{t-1}$ to $X_t$. When the index is odd (segments 2, 4), $(a_t, b_t) = (0.2, 0.7)$, indicating a dominant $X \to Y$ causal direction due to the strong influence from $X_{t-1}$ to $Y_t$. The dominant causal direction thus alternates between $Y \to X$ and $X \to Y$ across the five segments.

Fig. 2 illustrates the evolution of dynamic causality between $X_t$ and $Y_t$ as discovered by UnCLe and GVAR [16]. For better interpretability, the strengths are smoothed using a Gaussian moving average and presented as two lines. The orange and green segments on the timestamp axis divided by red dotted vertical lines indicate which variable is dominant over the other. UnCLe initially identifies $Y_t$ as determining $X_t$, flips the causal direction shortly after $t = 400$, and reverts to the another causal direction correctly after each subsequent switch points. This behavior aligns perfectly with the underlying data generation mechanism, demonstrating UnCLe's capability to accurately capture temporal causal dynamics. In contrast, while GVAR generates dynamic causal strength, the perceived dominance between $Y_t$ and $X_t$ never flips.

Table 2 quantifies the accuracy of dynamic causal discovery accuracy on TVSEM by evaluating separately on each segments with different system settings. Static Best denotes the best possible accuracy by a non-changing static causal graph. UnCLe(P) achieved perfect estimated of the directions of the two variables.

Table 2: Dynamic causal discovery performance comparison on TVSEM and ND8.

| Methods | TVSEM | | ND8 | |
|---|---|---|---|---|
| | AUROC ↑ | AUPRC ↑ | AUROC ↑ | AUPRC ↑ |
| GVAR | 0.733(±.000) | 0.400(±.000) | 0.723(±.016) | 0.220(±.028) |
| UnCLe(P) | **1.000**(±.000) | **1.000**(±.000) | **0.921**(±.007) | 0.633(±.045) |
| Static Best | 0.467(±.000) | 0.300(±.000) | 0.905(±.000) | **0.799**(±.000) |

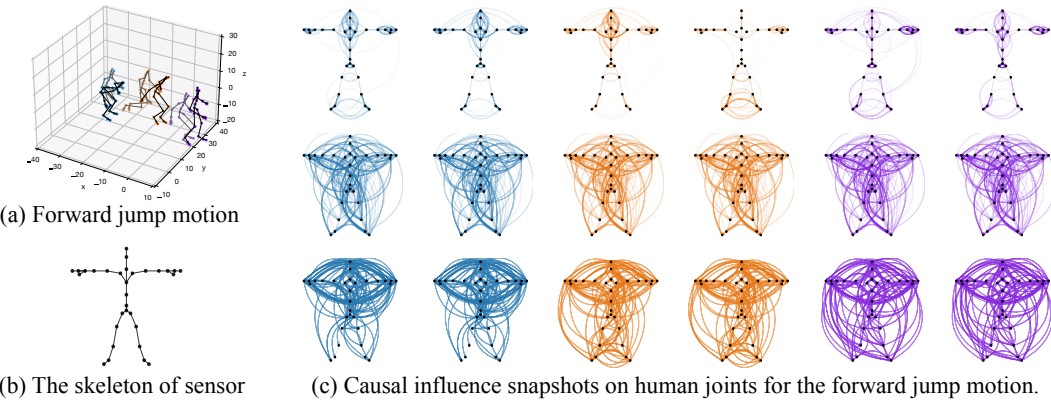

(a) Forward jump motion

(b) The skeleton of sensor positions on human joints

(c) Causal influence snapshots on human joints for the forward jump motion. From top to botton at each row: UnCLe, GVAR, JRNGC.

Figure 3: Dynamic causal analysis on a forward jump motion.

**Dynamic: ND8**  ND8 is a much harder dataset compared to TVSEM as it contains non-linear connections, more variables, multiple switches of direction simultaneously. The detailed data generated process of ND could be found in Appendix G. As reported in Table 2, UnCLe achieved better dynamic causal discovert accuracy then GVAR and Static Best.

**Dynamic: Human Motion Capture (MoCap)**  The Motion Capture (MoCap) dataset contains sensor observations reflecting the sophisticated biomechanics of the human body, which involve dynamic cooperation and interactions across joints, muscles, and bones. This dataset records the 3-axis angles of 31 joints. We selected a forward jump as a representative motion for our study, illustrated in Figure 3(a). The underlying joint skeleton is depicted in Figure 3(b).

To evaluate the ability of different methods to capture evolving causal structures, we extracted 6 snapshots of the causal graphs inferred by UnCLe, GVAR, and JRNGC from the forward jump data. For JRNGC, which does not inherently produce dynamic graphs, these snapshots were obtained by training the model on distinct segments of the motion data corresponding to different phases. These 6 snapshots are presented chronologically in Figure 3(c), aligned with the three motion phases (Crouch, Flight, Touchdown), and anchored to the skeletal structure for visual interpretation. The causal graphs generated by UnCLe demonstrate comparatively clear interpretability corresponding to the biomechanics of each phase:

*Crouch Phase* (first two UnCLe snapshots). Focus on the upper body, particularly the coordinated movement of the arms, with some connections to the lower body and strong links to the hip/root joint. This aligns with biomechanical findings that a coordinated arm swing is crucial for maximizing jump height by increasing the work and torque produced by the lower extremities. The dense, whole-body connectivity discovered by UnCLe reflects this principle of synergistic power generation for propulsion [8].

*Flight Phase* (middle two UnCLe snapshots). Highlights the lower body, characterized by coordinated leg movements, minimal upper body involvement, and weaker connections to the hip/root joint. This aligns with the biomechanical expectation that during mid-flight, with the body's trajectory already determined, the coordination strategy shifts from power generation to in-air balance. The graph correctly becomes sparser, reflecting a reduction in active, large-scale interdependencies.

*Touchdown Phase* (last two UnCLe snapshots). Reveals involvement from both upper and lower body segments with medium-strength connections, evidence of ipsilateral coordination, and renewed strong links to the hip/root joint. This is consistent with the demands of landing, which requires the entire kinetic chain—from the ankle up to the hip and core—to work in a coordinated fashion to absorb impact forces and re-stabilize the body. The re-emergence of a complex causal graph mirrors the body's need to manage ground reaction forces and dissipate energy across multiple joints [6].

In contrast, the snapshots generated by the baseline methods (GVAR and JRNGC) exhibit more subtle differences between phases. Their respective causal graphs often appear densely interconnected and are considerably more challenging to interpret in terms of distinct biomechanical phases.

To quantify the extent to which these methods recover fundamental anatomical connections, Table 3 reports the proportion of missing edges corresponding to adjacent joint connections present in the basic skeleton (Figure 3(b)) that were not captured in the inferred causal graph snapshots (averaged across the six snapshots for each method). UnCLe demonstrates a superior ability to preserve these fundamental T-pose connections, indicated by a lower missing rate. This experiment suggests that dynamic causal discovery algorithms like UnCLe hold significant promise for elucidating the mechanisms underlying real-world phenomena and complex systems by providing interpretable, time-evolving causal insights.

Table 3: Missing rate of skeletal connections.

| Method | Missing Rate ↓ |
|---|---|
| UnCLe | **.200**(±.019) |
| GVAR | .622(±.031) |
| JRNGC | .600(±.000) |

**Time Efficiency**   Figure 4 presents a comparative scatter plot of UnCLe(P) and baseline methods, illustrating their trade-off between causal discovery accuracy (AUROC) and computational time on the Lorenz#1 dataset. UnCLe demonstrates a compelling balance: it achieves the highest AUROC score while maintaining a competitive execution time. Specifically, UnCLe is notably faster than several complex neural methods such as TCDF and score-based methods like Dynotears, and exhibits comparable or moderately higher computational cost than some traditional or highly optimized approaches like VAR and CUTS+, respectively. This positions UnCLe as an effective and relatively efficient solution for accurate causal discovery.

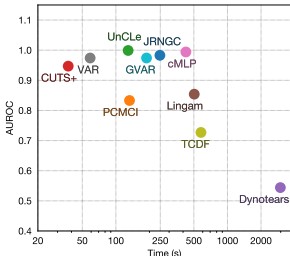

Figure 4: Time efficiency and causal discovery accuracy on Lorenz#1.

## 5   Ablation Study

We analyze the importance of UnCLe's key architectural components and methodological choices. First, we evaluate the contributions of *parameter sharing*, the *Auto-regressive Dependency Matrices*, and the *prediction task* using the high-dimensional Lorenz#3 dataset. Figure 5 presents the performance of the standard UnCLe(P) model compared to modified versions. Second, we assess the sensitivity of our causal discovery mechanism to different perturbation strategies on the Lorenz#1 dataset.

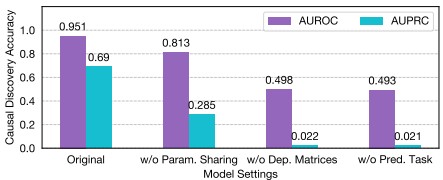

Figure 5: Ablation study on UnCLe's key components.

**w/o Parameter Sharing**   Disables the parameter sharing strategy, resulting in individual TCN Uncoupler and Recoupler pairs being trained for each time series component-wise. As reported, the causal discovery performance drops significantly, although this variation can still learn some valid causal structures. This outcome demonstrates the effectiveness of representational knowledge sharing, particularly in high-dimensional scenarios.

**w/o Auto-regressive Dependency Matrices**   Omit the Dependency Matrices of UnCLe. The parameter sharing strategy is also disabled. The multivariate time series prediction task is handled directly by the TCN Uncoupler and Recoupler. This leads to the AUROC score dropping below 0.5, equivalent to random guessing, indicating that the model can no longer effectively learn the causal structure. This result highlights the critical role of Dependency Matrices in uncoupling and explicitly capturing inter-variable dependencies in multivariate time series. Additionally, when the Dependency Matrices are disabled, causal structure inference via dependency aggregation becomes unavailable.

**w/o Prediction Task**   When the model is optimized solely for the reconstruction task (i.e., predicting $x^t$ using $x^{\le t}$), the causal structure cannot be effectively extracted, as the task becomes overly trivial. Each time series learns to reconstruct itself based solely on its own data rather than integrating information from others. This underscores the necessity of the prediction task as a bridge for modeling complex systems and learning inter-variable dependencies.

**Perturbation Strategies**   To validate our choice of temporal permutation, we compare its performance against three alternative strategies on the Lorenz#1 dataset: (1) *Zero-Masking*, where the target series is replaced with zeros; (2) *Noise Injection*, where Gaussian white noise is added to the target series; and (3) *No Perturbation*, which serves as a baseline to confirm that error gain is necessary. The results are shown in Table 4.

Table 4: Causal discovery performance (UnCLe(P)) on Lorenz#1 with different perturbation strategies.

| Perturbation Strategy | AUROC ↑ | AUPRC ↑ | ACC ↑ |
|---|---|---|---|
| Temporal Permutation (Ours) | **.999**(±.002) | **.996**(±.008) | **.994**(±.010) |
| Noise Injection | .981(±.056) | .946(±.134) | .978(±.048) |
| Zero-Masking | .974(±.082) | .932(±.177) | .969(±.052) |
| No Perturbation | .500(±.000) | .575(±.000) | .850(±.000) |

Temporal permutation significantly outperforms the alternatives. We reason that this is because it uniquely satisfies two crucial conditions: it effectively nullifies the predictive temporal information while perfectly preserving the variable's marginal distribution, thus ensuring model stability. In contrast, Zero-Masking disrupts the data distribution, and Noise Injection does not fully remove the original signal. The "No Perturbation" baseline confirms that without a valid perturbation, the method defaults to random guessing (AUROC $\approx 0.5$), validating the core principle of our post-hoc analysis.

# 6 Limitations and Future Work

The primary limitation of our work, which also defines a critical direction for future research, is the lack of formal identifiability guarantees. While UnCLe demonstrates strong empirical performance, we do not provide a theoretical proof under which conditions it is guaranteed to recover the true dynamic causal graph. Establishing the theoretical conditions under which our learned latent space provides a causally faithful linearization remains a key open question. Our future work will focus on bridging this gap, potentially by exploring connections to causal representation learning and imposing further structural constraints on the model to ensure that the learned latent dynamics are not just predictive, but verifiably causal.

# 7 Broader Impacts

**Potential for Misuse.**   As with any observational causal discovery method, the outputs of UnCLe are hypotheses subject to underlying assumptions (e.g., no hidden confounders) and should not be interpreted as definitive proof of causation. The primary risk lies in the uncritical application of our method in high-stakes domains, such as finance, healthcare, or social policy, where spurious causal claims could lead to flawed and potentially harmful decisions.

**Safeguards and Responsible Application.**   To mitigate these risks, we strongly advocate for responsible use. The causal graphs generated by UnCLe should be treated as a tool for exploration and hypothesis generation, not as a substitute for rigorous scientific validation. We recommend that any findings be validated by domain experts and, where feasible, tested through controlled experiments or prospective studies before being used for decision-making.

# 8 Conclusion

In this paper, we propose a novel dynamic causal discovery method, UnCLe, which consists of a pair of Uncouplers and Recouplers alongside Dependency Matrices. This architecture disentangles input time series into semantic representations and learns causal connections between variables through auto-regressive prediction. Extensive experiments demonstrate UnCLe's effectiveness and scalability across static and dynamic datasets from diverse domains. By bridging the gap in dynamic causal discovery methods, UnCLe aims to inspire further advancements in this domain.

## Acknowledgments and Disclosure of Funding

This work is partially supported by the National Natural Science Foundation of China (92167104, 62072006), CCF-Ant Research Fund, Qiyuan Lab Innovation Fund, and National Key Laboratory of Intelligent Parallel Technology.

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

# A Dataset Details

**Overview of Datasets** We evaluate UnCLe using datasets from a great variety of domains, as detailed in Table 5. The table provides information on the dataset types, the number of variables ($p$), the series length ($T$), and the number of replicas ($R$). The replicas of a dataset start with different initial system status. We provide the time series and true causal adjacency matrices in CSV format of all datasets in our public code and datasets repository.

Table 5: Used synthetic static (top), synthetic dynamic (middle) and real-world (bottom) datasets.

| Dataset | Type | $p$ | $T$ | $R$ |
|---|---|---|---|---|
| Lorenz#1 | climate dynamics ODE | 20 | 250 | 5 |
| Lorenz#2 | climate dynamics ODE | 20 | 250 | 5 |
| Lorenz#3 | climate dynamics ODE | 100 | 500 | 5 |
| fMRI | medical measurements | 15 | 200 | 50 |
| NC8 | nonlinear constant interactions | 8 | 2000 | 5 |
| FINANCE | financial portfolios | 20, 40 | 4000 | 8 |
| TVSEM | time-variant auto-regressive | 2 | 2000 | 5 |
| ND8 | nonlinear dynamic interactions | 8 | 2000 | 5 |
| MoCap | human motion capture | 93 | $\approx 300$ | 8 |
| METR-LA | traffic flow speed | 207 | 10240 | 1 |
| PEMS-BAY | traffic flow speed | 325 | 10240 | 1 |

## A.1 Synthetic Datasets

*Lorenz96* [14] is a nonlinear model to simulate climate dynamics used by [16, 27, 10]. The system dynamics become increasingly chaotic and thus hard to model for higher values of the forcing constant $F$. We design three sets of system configurations of Lorenz96, setting $p = \{20, 20, 100\}$, $T = \{250, 250, 500\}$, and $F = \{10, 40, 40\}$ for Lorenz#1, #2, and #3 respectively.

*fMRI* (functional Magnetic Resonance Imaging) [25] used by [16, 10, 17] contains time-ordered samples of the blood-oxygenation-level dependent (BOLD) signals, measuring activity in different brain regions of interest in human subjects.

*NC8* (Non-linear Constant interactions with $p = 8$ variables) is a dataset we propose that contains a wide variety of inter-variable interactions with time lags ranging from 1 (short-term) to 16 (long-term). The generating equations include non-linear functions such as $\sin(\cdot)$, $(\cdot)^3$, and $\max(\cdot)$, and involve all three common causal structures: fork, chain, and collision [21]. We provide the detailed generation equations in Appendix F.

*TVSEM* (Time-Varying Structural Equation Model) is a bivariate synthetic dataset we constructed to evaluate dynamic causal discovery. It features two variables whose causal dominance switches periodically every 400 timesteps over a total length of $T = 2000$, governed by changing autoregressive coefficients, as detailed in the main paper.

*ND8* (Non-linear Dynamic interaction with $p = 8$ variables) is the dynamic version of NC8, where some connections from the original dataset change direction periodically. We provide the detailed generation equations in Appendix G.

*FINANCE* [11] used in [17] is a simulated financial time series dataset that uses a factor model to describe a portfolio's return.

## A.2 Real-world Datasets

*MoCap* (CMU human motion capture) contains real-time 3-axis joint angles of 31 different parts of the human body at a frequency of 120 Hz. We selected seven actions from the database: walk, run, kick, jump, golf, sidestep, and bend.

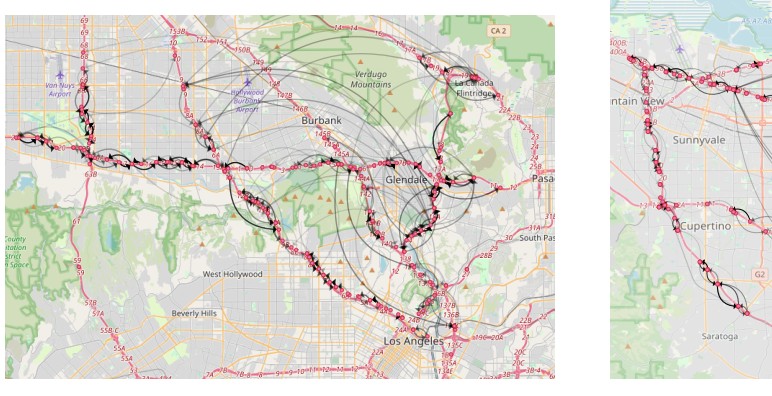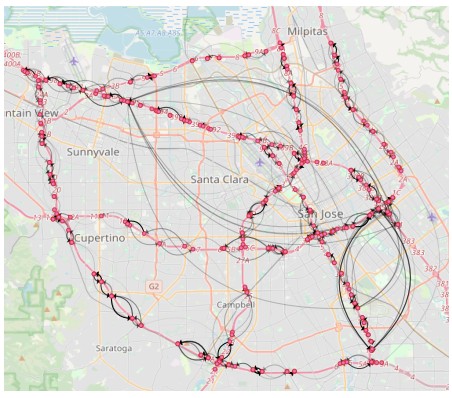

| (a) METR-LA by UnCLe | (b) PEMS-BAY by UnCLe |

Figure 6: Traffic roadmaps discovered by UnCLe in the METR-LA and PEMS-BAY datasets. The locations of the sensors are aligned with the map in the background.

*METR-LA* and *PEMS-BAY* [13] contain traffic speed data from highways in Los Angeles County and the Bay Area, respectively. The data is sampled at a 5-minute rate, and we use the first $T = 10240$ observations, spanning approximately 7 weeks.

## B  Methodology of Evaluation

All baseline methods produce weighted causal graphs. For the VAR method, we set $1 - p$ as strength of causal relationships where $p$ denotes the significance in Granger causality tests. We evaluate the causal discovery performance of all methods by comparing the inferred structures of against the true structure using areas under receiver operating characteristic (AUROC) and precision-recall (AUPRC) curves and accuracy (ACC). AUROC and AUPRC are measured on weighted graphs whereas ACC is calculated on binary adjacency matrices. We set the binarization thresholds to maximize ACC according the true causal structures. For all evaluation metrics, we only consider off-diagonal elements of adjacency matrices and ignore self-causal relationships which are basically always true and usually the easiest to infer . All reported metrics are the mean across all replications of the datasets, with 95% confidence intervals. The performance of the two available causal graph inference approaches of UnCLe are reported seperately, and we denote variable perturbation as *(P)* and weight aggregation as *(A)*. We perform grid search on the hyperparameters of all methods to maximize AUROC. We list the hyperparameter settings of UnCLe and other baseline methods in Section L.

## C  Additional Results on Large-scale Transportation Dataset (METR-LA and PEMS-BAY)

The METR-LA (207 sensors) and PEMS-BAY (325 sensors) datasets are large-scale transportation datasets. Using 10,240 data points, UnCLe and JRNGC attempt to recover the real-world road network from traffic flow speed sensor observations, as shown in Fig. 6 and 7.

In the graphs generated by UnCLe, most nodes are connected to their neighboring nodes, while the influence of a small set of hub nodes extends to distant areas. By referencing the maps, we identify these hub nodes as primarily airports or large overpasses, which handle the majority of traffic flow in their respective regions. The road network graphs discovered by UnCLe in real-world regions can be seamlessly integrated into practical scenarios, providing valuable support for analysis and decision-making.

In contrast, the graphs produced by JRNGC exhibit causal relationships scattered across the map, scarcely recovering connections between neighboring road network nodes. As a result, these graphs provide limited insights for real-world decision-making.

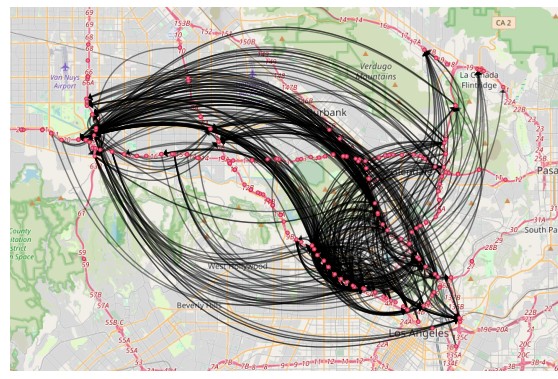
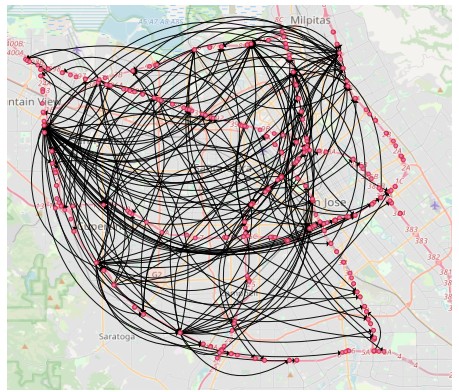

(a) METR-LA by JRNGC        (b) PEMS-BAY by JRNGC

Figure 7: Traffic roadmaps discovered JRNGC.

## D  Additional Results on ND8 with Static Baselines

To provide a comprehensive comparison on the ND8 dataset, which features dynamic ground truth causality, we also evaluate the performance of several static causal discovery baselines with the same hyperparameter settings as with NC8. This evaluation is presented in Table 6. Since these baseline methods inherently produce a single static causal graph, their output is assessed against the evolving ground truth of ND8. For context, the "Static Best" row in the table indicates the theoretical upper bound on performance achievable by any single, optimal static graph when evaluated against this dynamic ground truth. This effectively benchmarks how well any time-invariant model could possibly capture the changing causal relationships. As the results show, UnCLe(P), with its ability to model dynamic causality, significantly outperforms all evaluated static methods and surpasses the "Static Best" theoretical limit in terms of AUROC, highlighting the inherent advantage of dynamic approaches for such datasets.

Table 6: Causal discovery performance on the dynamic ND8 dataset. Static baselines are compared against the evolving ground truth, with "Static Best" representing the optimal performance for a single static graph.

| Methods | ND8 | |
|---|---|---|
| | AUROC ↑ | AUPRC ↑ |
| VAR | $0.578_{(\pm.035)}$ | $0.053_{(\pm.033)}$ |
| PCMCI | $0.848_{(\pm.028)}$ | $0.369_{(\pm.062)}$ |
| cMLP | $0.686_{(\pm.024)}$ | $0.152_{(\pm.045)}$ |
| TCDF | $0.741_{(\pm.011)}$ | $0.292_{(\pm.007)}$ |
| VARLiNGAM | $0.902_{(\pm.055)}$ | $0.614_{(\pm.129)}$ |
| DYNOTEARS | $0.533_{(\pm.001)}$ | $0.086_{(\pm.006)}$ |
| CUTS+ | $0.805_{(\pm.009)}$ | $0.345_{(\pm.014)}$ |
| JRNGC | $0.744_{(\pm.034)}$ | $0.151_{(\pm.011)}$ |
| GVAR | $0.723_{(\pm.016)}$ | $0.220_{(\pm.028)}$ |
| UnCLe(P) | $\mathbf{0.921}_{(\pm.007)}$ | $0.633_{(\pm.045)}$ |
| Static Best | $0.905_{(\pm.000)}$ | $\mathbf{0.799}_{(\pm.000)}$ |

## E  Additional Results on fMRI

We extended our evaluation of UnCLe to include fMRI dataset, a synthetic medical dataset. Performance metrics are detailed in Table 7. On this dataset, UnCLe(P) achieved a competitive Accuracy (ACC) of $0.925_{(\pm.010)}$, underscoring its ability to correctly classify the presence or absence of connections.

Table 7: Causal discovery performance on the fMRI dataset.

| Method | fMRI Dataset | | |
|--------|--------|--------|--------|
| | AUROC ↑ | AUPRC ↑ | ACC ↑ |
| VAR | $0.615_{(\pm.088)}$ | $0.175_{(\pm.108)}$ | $0.910_{(\pm.012)}$ |
| PCMCI | $\mathbf{0.813}_{(\pm.096)}$ | $0.278_{(\pm.156)}$ | $0.924_{(\pm.008)}$ |
| cMLP | $0.616_{(\pm.136)}$ | $0.191_{(\pm.116)}$ | $0.846_{(\pm.050)}$ |
| GVAR | $0.687_{(\pm.132)}$ | $\underline{0.289}_{(\pm.232)}$ | $0.806_{(\pm.140)}$ |
| TCDF | $\underline{0.812}_{(\pm.082)}$ | $\mathbf{0.368}_{(\pm.252)}$ | $0.899_{(\pm.046)}$ |
| VARLiNGAM | $0.677_{(\pm.131)}$ | $0.264_{(\pm.173)}$ | $0.924_{(\pm.009)}$ |
| DYNOTEARS | $0.544_{(\pm.055)}$ | $0.315_{(\pm.049)}$ | $0.857_{(\pm.008)}$ |
| CUTS+ | $0.689_{(\pm.116)}$ | $0.212_{(\pm.145)}$ | $0.924_{(\pm.009)}$ |
| JRNGC | $0.776_{(\pm.030)}$ | $0.289_{(\pm.066)}$ | $\mathbf{0.975}_{(\pm.001)}$ |
| UnCLe(P) | $0.792_{(\pm.118)}$ | $0.286_{(\pm.154)}$ | $\underline{0.925}_{(\pm.010)}$ |
| UnCLe(A) | $0.783_{(\pm.068)}$ | $0.235_{(\pm.108)}$ | $0.923_{(\pm.006)}$ |

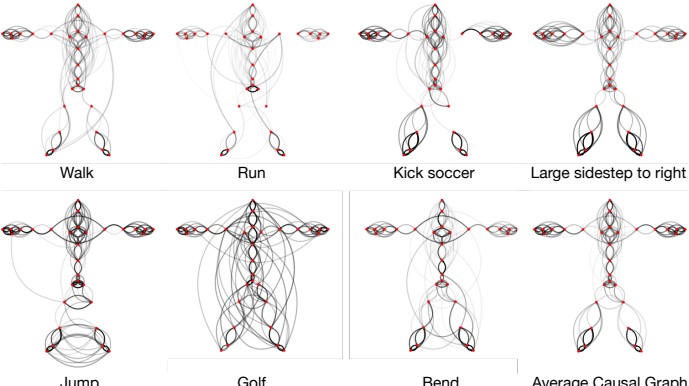

Figure 8: The discovered causal structures of human joints across 7 collected human actions. The last structure is averaged from these 7 actions.

## F   Additional Results on MoCap

We use UnCLe to analyze 7 different human actions, and the resulting causal structure is shown in Figure 8. Note that we aggregate the 3-axis variables to a single variable that represents the joint by max pooling to display the causal structure more clearly. Generally, the found causal structure is in line with our intuition on how joints of humans affect each other when we perform specific actions, to name a few: the structure of walking and running are similar; the causal structure is prominent around legs in soccer kicking and sidestepping; jump shows complex connections on feet; golf shows most sophisticated causal relation as this sport basically involves every muscle of the human body. In conclusion, UnCLe provides effective insight into how the joints in our physical body work collaboratively to complete motion actions.

Figure 9 shows the joint structure discovered from "kick soccer" motion of the MoCap dataset by UnCLe, VAR, PCMCI, cMLP and CUTS+. The results from UnCLe are the clearest and align more closely with the actual patterns of human motion. Additionally, UnCLe is capable of detecting differences in the strength of relationships, whereas the differences detected by other methods are very subtle.

## G   NC8: A Synthetic Dataset with Nonlinear Interactions

The time series of NC8 dataset are generated with following equations:

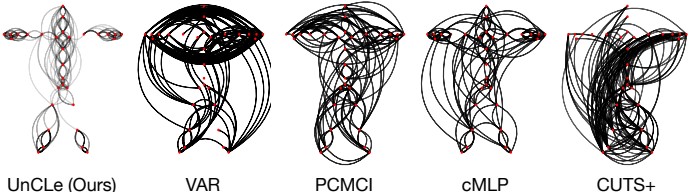

| UnCLe (Ours) | VAR | PCMCI | cMLP | CUTS+ |

Figure 9: The discovered causal structures of human joints of the "kick soccer" motion on UnCLe and other methods.

$$x_t = 0.45 \sin \frac{t}{4\pi} + 0.45 \sin \frac{t}{9\pi} + 0.25 \sin \frac{t}{3\pi} + 0.1\epsilon_{\mathbf{x}}$$

$$y_t = 0.24x_{t-1} - 0.28x_{t-2} + 0.08x_{t-3} + 0.2x_{t-4} +$$
$$0.2y_{t-1} - 0.12y_{t-2} + 0.16y_{t-3} + 0.04y_{t-4} + 0.02\epsilon_{\mathbf{y}}$$

$$z_t = 3 \cdot (0.6x_{t-1})^3 + 3 \cdot (0.4x_{t-2})^3 + 3 \cdot (0.2x_{t-3})^3 +$$
$$3 \cdot (0.5x_{t-4})^3 + 0.02\epsilon_{\mathbf{z}}$$

$$w_t = 0.8 \cdot (0.4z_{t-1})^3 + 0.8 \cdot (0.5z_{t-2})^3 + 0.64z_{t-3} +$$
$$0.48z_{t-4} + 0.02\epsilon_{\mathbf{w}}$$

$$a_t = 0.15 \sin \frac{t}{6} + 0.35 \sin \frac{t}{80} + 0.65 \sin \frac{t}{125} + 0.1\epsilon_{\mathbf{a}}$$

$$b_t = 0.54a_{t-13} - 0.63a_{t-14} + 0.18a_{t-15} + 0.45a_{t-16} +$$
$$0.36b_{t-13} + 0.27b_{t-14} - 0.36b_{t-15} + 0.18b_{t-16} + 0.02\epsilon_{\mathbf{b}}$$

$$c_t = \max(0.24a_{t-13} + 0.3a_{t-14}, -0.2) +$$
$$1.2\sqrt{|0.2a_{t-15} + 0.5x_{t-16}|} + 0.02\epsilon_{\mathbf{c}}$$

$$o_t = 0.39x_{t-13} - 0.65x_{t-14} + 0.52x_{t-15} + 0.13x_{t-16} +$$
$$0.52a_{t-1} - 0.65a_{t-2} + 0.26a_{t-3} + 0.52a_{t-4} + 0.02\epsilon_{\mathbf{o}}$$

where $\epsilon_{(\cdot)} \sim \mathcal{N}(0, 1)$ are the noise factors conforming to the standard normal distribution. The NC8 dataset contains 5 replicas with different random seeds and different $t_0 = [0, 100, 200, 300, 400]$ beginning offsets. Figure 10 illustrates the causal structure of NC8, with the numbers on the edges indicating the lags of influence.

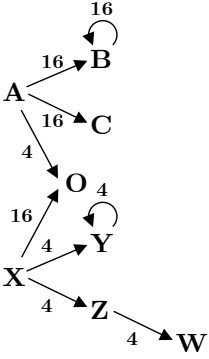

Figure 10: The true causal structure of NC8.

## H ND8: A Synthetic Dataset with Dynamic Causal Structures

The ND8 dataset is designed to evaluate the capability of methods to detect dynamic, or time-varying, causal relationships. It is derived from the static NC8 dataset by introducing periodic switches in

the causal dependencies between specific variable pairs. Specifically, the generating equations for variables $z_t, w_t, c_t$, and $o_t$ remain identical to those defined for the NC8 dataset (as presented in the previous section). However, the causal relationships involving variables $x_t, y_t, a_t$, and $b_t$ are subject to change.

The dynamic nature is implemented as follows: the primary causal direction between the pair $(x, y)$ and the pair $(a, b)$ reverses every 500 timesteps. Initially (e.g., for timesteps $t = 1, \ldots, 500; 1001, \ldots, 1500;$ etc.), the generating equations for $x_t, y_t, a_t$, and $b_t$ are:

$$
\begin{aligned}
x_t =&\, 0.45 \sin \frac{t}{4\pi} + 0.45 \sin \frac{t}{9\pi} + 0.25 \sin \frac{t}{3\pi} + 0.1\epsilon_{\mathbf{x}} \\
y_t =&\, 0.24x_{t-1} - 0.28x_{t-2} + 0.08x_{t-3} + 0.2x_{t-4} + \\
&\, 0.2y_{t-1} - 0.12y_{t-2} + 0.16y_{t-3} + 0.04y_{t-4} + 0.02\epsilon_{\mathbf{y}} \\
a_t =&\, 0.15 \sin \frac{t}{6} + 0.35 \sin \frac{t}{80} + 0.65 \sin \frac{t}{125} + 0.1\epsilon_{\mathbf{a}} \\
b_t =&\, 0.54a_{t-13} - 0.63a_{t-14} + 0.18a_{t-15} + 0.45a_{t-16} + \\
&\, 0.36b_{t-13} + 0.27b_{t-14} - 0.36b_{t-15} + 0.18b_{t-16} + 0.02\epsilon_{\mathbf{b}}
\end{aligned}
$$

During the alternate 500-timestep intervals (e.g., for timesteps $t = 501, \ldots, 1000; 1501, \ldots, 2000;$ etc.), the generating equations for $x_t, y_t, a_t, b_t$ switch to the following:

$$
\begin{aligned}
x_t =&\, 0.08x_{t-1} - 0.08x_{t-2} + 0.04x_{t-3} + 0.04x_{t-4} + \\
&\, 0.04y_{t-1} + 0.28y_{t-2} - 0.08y_{t-3} - 0.04y_{t-4} + 0.1\epsilon_{\mathbf{x}} \\
y_t =&\, 0.45 \sin \frac{t}{4\pi} + 0.45 \sin \frac{t}{9\pi} + 0.25 \sin \frac{t}{3\pi} + \\
&\, 0.2y_{t-1} - 0.12y_{t-2} + 0.16y_{t-3} + 0.04y_{t-4} + 0.02\epsilon_{\mathbf{y}} \\
a_t =&\, 0.09a_{t-13} - 0.18a_{t-14} + 0.09a_{t-15} + 0.09a_{t-16} + \\
&\, 0.72b_{t-13} + 0.27a_{t-14} - 0.63a_{t-15} + 0.18a_{t-16} + 0.1\epsilon_{\mathbf{a}} \\
b_t =&\, 0.15 \sin \frac{t}{6} + 0.35 \sin \frac{t}{80} + 0.65 \sin \frac{t}{125} + \\
&\, 0.36b_{t-13} + 0.27b_{t-14} - 0.36b_{t-15} + 0.18b_{t-16} + 0.02\epsilon_{\mathbf{b}}
\end{aligned}
$$

In all equations, $\epsilon_{(.)} \sim \mathcal{N}(0, 1)$ represent independent noise factors drawn from a standard normal distribution. The ND8 dataset comprises 5 replicas, each generated with different random seeds for the noise terms. The evolving ground truth causal structure of the ND8 dataset is illustrated in Figure 11, with Figure 11a showing the initial causal relationships and Figure 11b depicting the structure after the causal switches. To provide a concrete visualization of the generated time series, Figure 12 displays one such replica from the ND8 dataset, generated using random seed 500. The plot clearly demarcates the "Reversal Points" at $t = 500, 1000, 1500$, where the causal dependencies between specific variable pairs ($x - y$ and $a - b$) are designed to switch.

# I   Interpreting Channel-wise Causal Contributions

Figure 13 provides a visual inspection of the learned Dependency Matrices, $\mathbf{\Psi}^c$, for each of the $C = 20$ semantic channels after training UnCLe on a synthetic dataset with a known ground truth causal graph $\mathcal{G}$. Each matrix $\mathbf{\Psi}^c$ (displayed as $\mathbf{\Psi}^0$ through $\mathbf{\Psi}^{19}$ in the figure) represents the inter-variable dependencies captured within that specific channel.

As observed in Figure 13, the individual channel-wise dependency matrices exhibit varied structures. Some channels (e.g., channels 0-3, 7-15, 17) learn very sparse connections, suggesting they might focus on noise modeling or highly specific, subtle interactions not prominent in the overall ground truth. Other channels, however, capture more discernible patterns of dependency. For instance, channel 19 appears to strongly reflect the primary diagonal dependencies present in $\mathcal{G}$, while channels such as 6 and 16 seem to contribute to capturing off-diagonal interactions. Channels 4 and 5 also highlight certain non-diagonal relationships. This visualization suggests that different channels

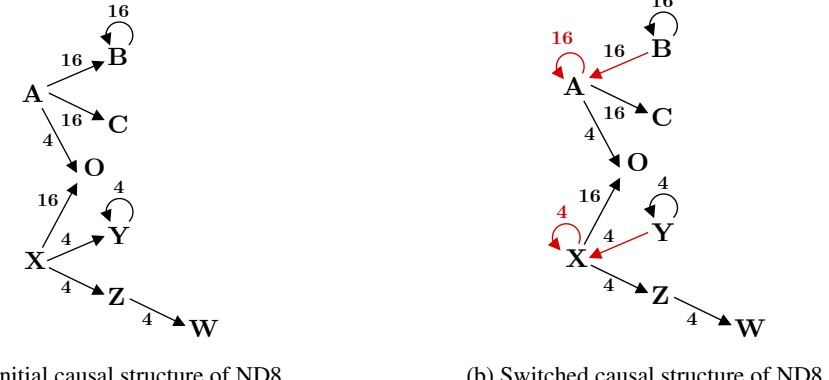

(a) Initial causal structure of ND8

(b) Switched causal structure of ND8

Figure 11: The true dynamic causal structure of the ND8 dataset, illustrating the initial state and the state after causal relationship reversals.

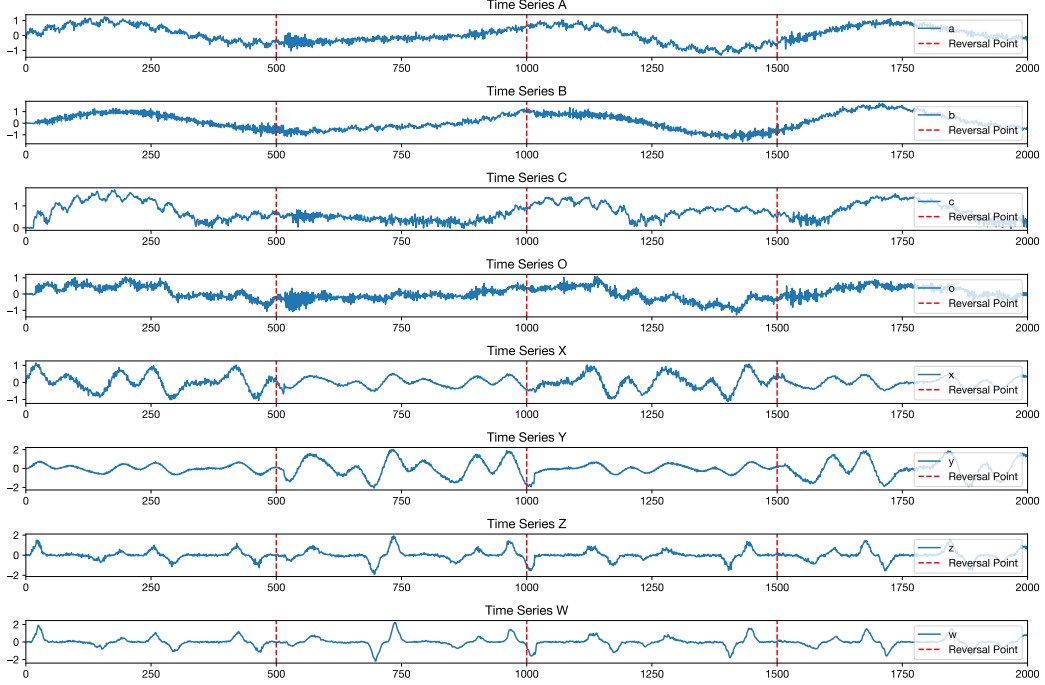

Figure 12: Visualization of a single replica from the ND8 dataset (generated with seed 500). The vertical dashed lines indicate the "Reversal Points" at $t = 500, 1000, 1500$, where predefined causal relationships switch.

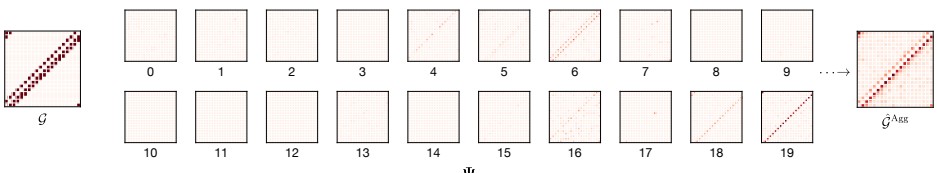

Figure 13: The full demostration of UnCLe's causal discovery via weight aggregation. $\mathcal{G}$ is the true causal graph and $\hat{\mathcal{G}}^{\text{Agg}}$ is the aggregated causal graph by averaging all $\boldsymbol{\Psi}$ from $\boldsymbol{\Psi}^0$ to $\boldsymbol{\Psi}^{19}$.

indeed learn to emphasize different facets or subsets of the underlying systemic dependencies, rather than each channel necessarily isolating entirely distinct and orthogonal causal mechanisms.

The final aggregated static causal graph, $\hat{\mathcal{G}}^{\text{Agg}}$, is derived by pooling information across all these channel-specific dependency matrices (as described in Section 3.2 on Static Causal Graph via Dependency Aggregation). The resulting $\hat{\mathcal{G}}^{\text{Agg}}$ in Figure 13 demonstrates a close resemblance to the true causal graph $\mathcal{G}$. This illustrates how the aggregation of these diverse, channel-specific perspectives allows UnCLe to reconstruct a comprehensive and accurate representation of the overall static causal structure, even if individual channels provide only partial or specialized views.

## J  Demonstration of UnCLe's Causal Discovery Mechanisms

UnCLe offers two primary mechanisms for causal discovery, as outlined in Section 3.2 Post-hoc Causal Discovery: (P) dynamic causal graph inference via temporal perturbation and analysis of datapoint-wise prediction errors, and (A) static causal graph inference via the aggregation of learned Dependency Matrices. We illustrate these mechanisms below.

### J.1  Dynamic Causal Discovery via Temporal Perturbation

UnCLe's primary approach for dynamic causal discovery involves quantifying the impact of temporal perturbations on prediction accuracy (Section 3.2 Perturbation-based Dynamic Granger Causality). The principle is that if variable $x_j$ causally influences $x_i$, then disrupting the temporal information in $x_j$ (e.g., via permutation) should lead to a noticeable increase in the prediction error for $x_i$.

Figure 14 illustrates this concept. Consider the task of predicting series $x_9$. The left panel shows the model's predictions for $x_9$ under normal conditions (blue line) versus when series $x_8$ is perturbed (orange line). Assuming $x_8$ is a true cause of $x_9$ (as suggested by a typical Lorenz system structure or the ground truth $\mathcal{G}$ in Figure 15), perturbing $x_8$ significantly degrades the prediction quality for $x_9$, causing the orange line to deviate markedly from the blue line. This deviation, quantified as the datapoint-wise error gain $\Delta\epsilon_{9,t}^{\backslash 8}$, indicates a causal link from $x_8$ to $x_9$.

Conversely, the right panel of Figure 14 shows the predictions for $x_9$ when a non-causal (or weakly causal) variable, say $x_{12}$, is perturbed. In this case, the predictions with $x_{12}$ perturbed (orange line) remain very close to the original predictions (blue line). The minimal error gain $\Delta\epsilon_{9,t}^{\backslash 12}$ suggests a weak or absent causal link from $x_{12}$ to $x_9$.

By systematically applying such perturbations and quantifying the error gains for all variable pairs across all timesteps, UnCLe constructs the dynamic causal graph $\hat{\mathcal{G}}^{\text{Pert}}$. The heatmap on the far right of Figure 14 represents a static summary or snapshot derived from these dynamic causal influences, demonstrating how this perturbation-based analysis reveals the causal structure.

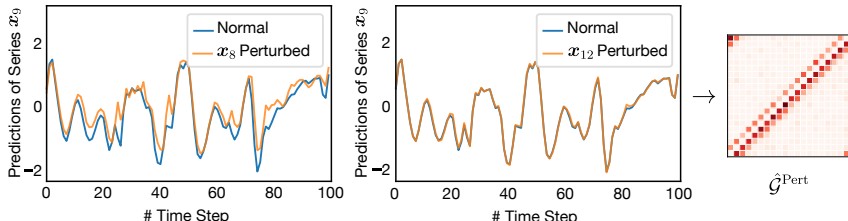

Figure 14: Demonstration of inferring causal influences for $\hat{\mathcal{G}}^{\text{Pert}}$ via temporal perturbation. Left: Predictions for $x_9$ with original data (Normal) vs. $x_8$ perturbed. Middle: Predictions for $x_9$ with original data vs. $x_{12}$ perturbed. Right: Resulting (static summary of) causal graph $\hat{\mathcal{G}}^{\text{Pert}}$ derived from such perturbation analysis.

### J.2  Static Causal Discovery via Dependency Matrix Aggregation

One approach UnCLe employs for static causal discovery is the aggregation of its learned Dependency Matrices ($\Psi$). As detailed in Section 3.2 Static Causal Graph via Dependency Aggregation and visualized in Figure 13, UnCLe learns multiple channel-specific Dependency Matrices, $\Psi^c$, each

capturing different aspects of inter-variable relationships. By pooling the information from all these channels (e.g., using the L2-norm), UnCLe constructs a single, comprehensive static causal graph, $\hat{\mathcal{G}}^{\text{Agg}}$. The effectiveness of this aggregation in accurately recovering the underlying causal structure (compared to a ground truth $\mathcal{G}$) is demonstrated in Figure 13, where $\hat{\mathcal{G}}^{\text{Agg}}$ closely matches $\mathcal{G}$ for the Lorenz#1 example. This method provides a direct way to obtain a summary causal graph from the trained model parameters.

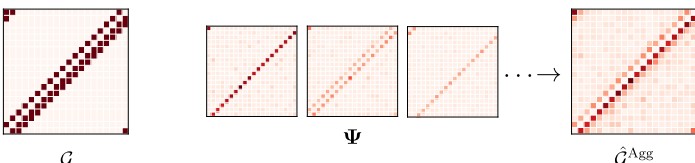

Figure 15: Demonstration of inferring the static causal graph $\hat{\mathcal{G}}^{\text{Agg}}$ via aggregation of Dependency Matrices $\Psi$. Left: True causal graph $\mathcal{G}$. Middle: Examples of learned Dependency Matrices (e.g., $\Psi^6, \Psi^{18}, \Psi^{19}$). Right: Aggregated causal graph $\hat{\mathcal{G}}^{\text{Agg}}$.

## K   Additional Ablation Study on Higher-Order Lags

To address the model's sensitivity to the autoregressive lag, we conducted an additional ablation study on the Lorenz#1 dataset. The 'lag' hyperparameter determines the temporal lookback for the linear prediction step in the latent space (Equation 3, modified to use lags > 1). The results for different lag values are presented in Table 8.

As shown, increasing the lag from 1 to 2 and 4 led to a decrease in causal discovery performance for both UnCLe(P) and UnCLe(A). We hypothesize this is because the TCN architecture's large receptive field already encodes sufficient long-range historical information into the latent representation $z_t$. Adding explicit higher-order lags in the linear prediction step may introduce parameter redundancy, making the model more prone to overfitting on spurious, non-primary relationships, while also increasing computational cost. This result suggests that a lag of 1 is sufficient and optimal for UnCLe's architecture on this task.

Table 8: Causal discovery performance on Lorenz#1 with different lag settings.

| Lag | UnCLe(P) | | UnCLe(A) | |
|---|---|---|---|---|
| | AUROC ↑ | AUPRC ↑ | AUROC ↑ | AUPRC ↑ |
| 1 | **.999**(±.002) | **.996**(±.008) | **.994**(±.007) | **.962**(±.054) |
| 2 | .961(±.042) | .878(±.089) | .956(±.021) | .822(±.071) |
| 4 | .877(±.076) | .670(±.121) | .940(±.041) | .783(±.099) |

## L   Hyperparameter Settings

Here we list the hyperparameter settings on all experiments method by method. For VAR, cMLP, TCDF, GVAR on the fMRI dataset, we adopt the experimental results of these methods in [16] and show the hyperparameter settings provided in that paper.

**UnCLe**   Table 9 lists the hyperparameter settings used for UnCLe of all experiments in this paper.

**cMLP** [27]   (available at https://github.com/iancovert/Neural-GC) Table 10 lists the hyperparameter settings used for cMLP. We use Hierarchical lasso as the sparsity penalty and run a 5-step grid search on the penalty factor $\lambda$.

**TCDF** [17]   (available at https://github.com/M-Nauta/TCDF) Table 11 lists the hyperparameter settings used for TCDF. We run a 5-step grid search on the significance level $\alpha$ of the Permutation Importance Validation Method.

Table 9: Hyperparameter Settings for UnCLe.

| Dataset | Lag | Kernel Size | TCN Blocks | Kernel Filters | Recon. Epochs | Joint Epochs | Learning Rate |
|---------|-----|-------------|------------|----------------|---------------|--------------|---------------|
| Lorenz#1 | 1 | 8 | 6 | 20 | 1,000 | 2,000 | 5e-3 |
| Lorenz#2 | 1 | 6 | 8 | 12 | 1,000 | 2,500 | 2e-3 |
| Lorenz#3 | 1 | 3 | 6 | 18 | 500 | 2,500 | 1e-3 |
| fMRI | 1 | 6 | 8 | 12 | 1,000 | 2,000 | 1e-5 |
| NC8 | 1 | 8 | 6 | 20 | 1,000 | 2,000 | 3e-4 |
| FINANCE | 2 | 2 | 3 | 24 | 500 | 10,000 | 3e-4 |
| ND8 | 1 | 8 | 6 | 20 | 1,000 | 2,000 | 3e-4 |
| TVSEM | 1 | 3 | 4 | 8 | 500 | 2,500 | 2e-3 |

Table 10: Hyperparameter Settings for cMLP.

| Dataset | Lag | Hidden Layers | Training Epochs | Learning Rate | Sparsity Hyperparams |
|---------|-----|---------------|-----------------|---------------|----------------------|
| Lorenz#1 | 5 | 1 | 2,000 | 1e-2 | $\lambda \in [0.0, 2.0]$ |
| Lorenz#2 | 5 | 1 | 3,000 | 1e-2 | $\lambda \in [0.0, 2.0]$ |
| Lorenz#3 | 5 | 1 | 2,000 | 1e-2 | $\lambda \in [0.0, 2.0]$ |
| fMRI | 1 | 1 | 2,000 | 1e-2 | $\lambda \in [1e-3, 0.75]$ |
| NC8 | 16 | 1 | 1,000 | 5e-3 | $\lambda \in [0.0, 2.0]$ |
| FINANCE | 3 | 1 | 1,000 | 1e-3 | $\lambda \in [0.0, 2.0]$ |

Table 11: Hyperparameter Settings for TCDF.

| Dataset | Kernel Size | Hidden Layers | Training Epochs | Learning Rate | Sparsity Hyperparams |
|---------|-------------|---------------|-----------------|---------------|----------------------|
| Lorenz#1 | 5 | 1 | 2,000 | 1e-2 | $\alpha \in [0.0, 2.0]$ |
| Lorenz#2 | 5 | 1 | 2,000 | 1e-2 | $\alpha \in [0.0, 2.0]$ |
| Lorenz#3 | 5 | 1 | 2,000 | 1e-2 | $\alpha \in [0.0, 2.0]$ |
| fMRI | 1 | 1 | 2,000 | 1e-3 | $\alpha \in [0.0, 2.0]$ |
| NC8 | 16 | 1 | 1,000 | 5e-3 | $\alpha \in [0.0, 2.0]$ |
| FINANCE | 5 | 1 | 2,000 | 1e-2 | $\alpha \in [0.0, 2.0]$ |

**GVAR** [16]  (available at https://github.com/i6092467/GVAR) Table 12 lists the hyperparameter settings used for GVAR. We run a 5x5-step grid search on the regularisation parameters $\lambda, \gamma$.

Table 12: Hyperparameter Settings for GVAR.

| Dataset | Lag | Hidden Layers | Training Epochs | Learning Rate | Sparsity Hyperparams |
|---------|-----|---------------|-----------------|---------------|----------------------|
| Lorenz#1 | 5 | 2 | 1,000 | 1e-4 | $\lambda \in [0.0, 3.0], \gamma \in [0.0, 0.025]$ |
| Lorenz#2 | 5 | 2 | 1,000 | 1e-4 | $\lambda \in [0.0, 3.0], \gamma \in [0.0, 0.025]$ |
| Lorenz#3 | 5 | 2 | 1,000 | 1e-4 | $\lambda \in [0.0, 3.0], \gamma \in [0.0, 0.025]$ |
| fMRI | 1 | 1 | 1,000 | 1e-3 | $\lambda \in [0.0, 3.0], \gamma \in [0.0, 0.1]$ |
| NC8 | 16 | 1 | 1,000 | 1e-4 | $\lambda \in [0.0, 3.0], \gamma \in [0.0, 0.025]$ |
| FINANCE | 3 | 2 | 500 | 1e-4 | $\lambda \in [0.0, 3.0], \gamma \in [0.0, 0.025]$ |

**VAR** [7]  (as implemented in the `statsmodels` library [23]) **& PCMCI** [22] (available at https://github.com/jakobrunge/tigramite))

VAR and PCMCI share the lag hyperparameter $L$. We set $L = 5$ on all the Lorenz96 experiments and the FINANCE experiment, $L = 1$ on fMRI, and $L = 16$ on NC8. The significance level of the PC algorithm of PCMCI is set to $0.01$.

---

**Algorithm 1** Causal discovery via temporal perturbation

---

**Input**: dataset $\boldsymbol{x}$; trained UnCLe model $f$.

**Output**: Adjacency matrix $\hat{\boldsymbol{A}}^{\text{Pert}}$.

1: $\hat{\boldsymbol{x}}_{2:T+1} \leftarrow f(\boldsymbol{x}_{1:T})$ {Predict on original dataset}
2: $\hat{\boldsymbol{A}}^{\text{Pert}} \leftarrow \boldsymbol{0}_{M \times M}$
3: **for** $i = 1$ to $N$ **do**
4:     $\epsilon_i = \ell(\hat{\boldsymbol{x}}_{i,2:T}, \boldsymbol{x}_{i,2:T})$ {Original error of series i}
5:     $\boldsymbol{x}^{\backslash i} \leftarrow \boldsymbol{x}$ {Clone the dataset}
6:     Permutate $\boldsymbol{x}_i^{\backslash i}$ {Perturb series i with permutation}
7:     $\hat{\boldsymbol{x}}^{\backslash i} \leftarrow f(\boldsymbol{x}^{\backslash i})$ {Predict on perturbed dataset}
8:     **for** $j = 1$ to $N$ **do**
9:         $\epsilon_i^{\backslash j} = \ell(\hat{\boldsymbol{x}}_{i,2:T}^{\backslash j}, \boldsymbol{x}_{i,2:T})$ {Perturbed error}
10:       **for** $t = 2$ to $T$ **do**
11:           $\Delta\epsilon_{i,t}^{\backslash j} = \max(0, \epsilon_{i,t}^{\backslash j} - \epsilon_{i,t})$
12:           $\hat{\boldsymbol{A}}_{j,i}^{t,\text{Pert}} \leftarrow \Delta\epsilon_{i,t}^{\backslash j}$ {Datapoint-wise error gain}
13:       **end for**
14:     **end for**
15: **end for**

---

**VARLiNGAM** [9]   (available at `https://github.com/cdt15/lingam`) We run a 4-step grid search on the lag from 2 to 5.

**DYNOTEARS** [19]   (available at `https://github.com/mckinsey/causalnex`) We set the max iteration to 1,000, regularisation parameters $\lambda_w$ and $\lambda_a$ to 0.1. We run a 4-step grid search on the lag from 2 to 5.

**CUTS+** [5]   (available at `https://github.com/jarrycyx/UNN/tree/main/CUTS_Plus`) We set learning rate to 1e-3, number of training epochs to 64 and max number of groups to 32. We run a 4-step grid search on the regularisation parameters $\lambda$ from 0.1 to 0.005.

**JRNGC** [30]   (available at `https://github.com/ElleZWQ/JRNGC`) We set the hidden size to 100, lag to 5, number of residual layers to 5 and learning rate to 1e-3. We run a 4-step grid search on the Jacobian regularizer coefficient $\lambda$ from 0.001 to 0.0001.

## M   Implementation of Temporal Perturbation

Algorithm 1 outlines the procedure for inferring the dynamic causal graph $\hat{\mathcal{G}}^{\text{Pert}}$ using temporal perturbation and datapoint-wise prediction errors. In this algorithm, $\ell$ denotes the Mean Squared Error (MSE) loss function, and $\boldsymbol{x}^{\backslash j}$ represents the dataset where the $j$-th time series, $\boldsymbol{x}_j$, has been perturbed (by permuting its temporal values). The core idea is to quantify the causal influence from variable $\boldsymbol{x}_j$ to variable $\boldsymbol{x}_i$ at time $t$. This is achieved by computing $\Delta\epsilon_{i,t}^{\backslash j}$, the datapoint-wise gain in prediction error for $\boldsymbol{x}_{i,t}$ when the historical information of $\boldsymbol{x}_j$ (i.e., $\boldsymbol{x}_{j,<t}$) is disrupted by perturbation. This error gain, $\Delta\epsilon_{i,t}^{\backslash j}$, serves as the strength of the causal link $\hat{\mathcal{G}}_{j,i}^{t,\text{Pert}}$ in the dynamic graph.

The computational efficiency of this process can be significantly enhanced through batch processing. Instead of perturbing and predicting for each series sequentially, we can prepare multiple perturbed versions of the dataset (each with a different series $\boldsymbol{x}_j$ perturbed) and process them in batches. By feeding these batches into the trained UnCLe model, predictions for multiple perturbed scenarios can be obtained in parallel. This optimization reduces the number of sequential forward passes through the model from $N$ (where $N$ is the number of series) to approximately $N/B$, where $B$ is the batch size, thereby reducing the overall inference time. The effective time complexity for the perturbation analysis, originally proportional to $\mathcal{O}(N \cdot T_{\text{model}})$, where $T_{\text{model}}$ is the time for one forward pass, becomes closer to $\mathcal{O}(\lceil N/B \rceil \cdot T_{\text{model}})$, assuming efficient parallelization within each batch.

