# OpenReview forum: "UnCLe: Towards Scalable Dynamic Causal Discovery in Non-linear Temporal Systems"
_NeurIPS.cc/2025/Conference — NeurIPS 2025 poster_

### Official Review · Reviewer_9Mey · 2025-06-20

**Clarity:** 3
**Significance:** 3
**Originality:** 3
**Rating:** 4
**Confidence:** 5

**Summary:**

This paper proposes the UnCLe algorithm, a Granger causal discovery method that can extract dynamic causal structures. The UnCLe algorithm first uses TCN to encode the original variable into a latent representation, and then searches for a dynamic causal dependency matrix in the latent space. It should be noted that a reconstruction loss supervises the latent representation.

**Questions:**

1. How can we ensure the identifiability of the causal dependency structure discovered by the proposed method in latent space? I will be happy to improve the score if this issue is solved.

2. The dynamic causal dependency matrix is ​​defined as $C$ learnable $N \times N$ matrices. This may cause problems when dealing with high-dimensional time series, as it significantly increases the number of parameters that need to be learned.

3. The learning strategy of Ψ seems to need further explanation. Is it a randomly initialized matrix? Is there any additional design?

**Ethical Concerns:**

["NO or VERY MINOR ethics concerns only"]

**Limitations:**

yes

**Quality:**

3

**Strengths And Weaknesses:**

**Strengths**

1. This method proposes a dynamic inter-slice causal structure algorithm based on Granger causality, which better matches the real dynamic system.

2. This method can obtain the causal dependencies of dynamic and static structures at the same time.

**Weaknesses**

1. Eq. (3) and Eq. (4) are Granger causal models that model lag 1, and it seems that the possibility of extending the model to higher orders is not explored.

2. The author provides a reasonable modeling method and mathematical expressions, but there is no theoretical analysis of the identifiability of cause and effect, especially since this article will learn causal relationships in the latent space.

---

> ### Author Rebuttal · Authors · 2025-07-31
>
> ### **Response to Reviewer 9Mey**
>
> We thank you for your time and for the valuable questions, which help us clarify important aspects of our method.
>
> **Weaknesses:**
>
> *   **W1: On not exploring higher-order lags.**
>     This is an excellent point. We conducted an ablation study on the Lorenz#1 dataset to investigate the effect of increasing the autoregressive lag in the Dependency Matrices. The results are as follows:
>
> | lag | AUROC (P) | AUPRC (P) | AUROC (A) | AUPRC (A) |
> |:-:|:---------------:|:---------------:|:---------------:|:---------------:|
> | 1 | **.999**($\pm$.002) | **.996**($\pm$.008) | **.994**($\pm$.007) | **.962**($\pm$.054) |
> | 2 | .961($\pm$.042) | .878($\pm$.089) | .956($\pm$.021) | .822($\pm$.071) |
> | 4 | .877($\pm$.076) | .670($\pm$.121) | .940($\pm$.041) | .783($\pm$.099) |
>
>
> Increasing the lag decreased performance. We hypothesize this is because the TCN's large receptive field already encodes long-range historical information into the latent representation $z_t$. Adding explicit higher-order lags in the linear prediction step introduces parameter redundancy, making the model more prone to overfitting on spurious, non-primary relationships, while also increasing computational cost. Thank you for raising this; we will add this experiment to the final version.
>
> **Questions:**
>
> *   **Q1: On the identifiability of the causal structure in latent space.**
>     (This is a core question also raised by Reviewers SUxp and T7tD). We agree that this is a critical theoretical point. Please see our detailed response to **Reviewer SUxp, Weakness W1**. In summary, while our work does not provide a formal proof of identifiability, which remains an open challenge for many deep learning methods, we provide extensive empirical evidence for UnCLe's effectiveness and consider this a key direction for future theoretical work.
>
> *   **Q2: On the parameter size of the Dependency Matrices in high-dimensional settings.**
>     The parameter count of the Dependency Matrices does grow quadratically with the number of variables $N$ (as $C \times N \times N$). However, the number of channels $C$ is a small, constant hyperparameter. More importantly, UnCLe's architecture is highly parallelizable. In cases where memory is a constraint on very large-scale data, the batch size $B$ can be reduced to allow the model to run, trading a small amount of speed for feasibility. This is detailed further in Appendix L ("Implementation of Temporal Perturbation").
>
> *   **Q3: On the learning strategy for the Dependency Matrices ($\Psi$).**
>     Our apologies for not detailing this sufficiently. The Dependency Matrices $\Psi$ are learnable parameters that are optimized jointly with the rest of the network. They are initialized from a normal distribution $N(0, \sqrt{3/N})$, where $N$ is the number of time series. Their values are updated based on their role in the prediction task (via the $L_{Pred}$ loss) and are encouraged towards sparsity by the $L_{L1}$ regularization term. We will add these details to the methodology section in the final version.
>
> Thank you again for your insightful questions and feedback.

---

> > ### Comment · Reviewer_9Mey · 2025-08-04
> >
> > Thank you very much for the detailed response. Considering that the author still did not fully address the questions I raised, but it is indeed difficult to give a theoretically recognizable result, I choose to keep my score unchanged.

---

### Official Review · Reviewer_T7tD · 2025-06-25

**Clarity:** 3
**Significance:** 3
**Originality:** 3
**Rating:** 5
**Confidence:** 3

**Summary:**

This paper introduces UnCLe (UnCoupLing causality), a novel deep learning-based framework for scalable, dynamic causal discovery in multivariate time series data. Unlike existing methods that primarily produce static or time-aggregated causal graphs, UnCLe explicitly models evolving temporal causality using a two-part architecture: (1) Uncoupler and Recoupler networks that disentangle input signals into semantic latent representations, and (2) auto-regressive Dependency Matrices to capture inter-variable dependencies. Dynamic causal influence is inferred post hoc by perturbing individual time series and measuring changes in prediction error, thus revealing time-specific causal structures. The authors support their claims with extensive experiments on synthetic, semi-synthetic, and real-world datasets, including human motion capture, and demonstrate that UnCLe outperforms state-of-the-art baselines in both accuracy and scalability.

**Questions:**

Can the authors clarify how the Dependency Matrices Ψ relate to true causal effects across different channels?

The causal strength estimation relies on random permutations of the input time series. How sensitive is UnCLe’s performance to the specific choice of perturbation strategy?

Since the method estimates causality by measuring increases in prediction error under perturbation, how does it guard against attributing causality to spurious correlations, especially in high-dimensional settings with limited data?

While UnCLe scales better than several baselines, can the authors provide more detailed runtime and memory usage analysis? This would help in assessing suitability for very high-dimensional settings.

**Ethical Concerns:**

["NO or VERY MINOR ethics concerns only"]

**Final Justification:**

After reading the rebuttal and the reply of the author(s), my score remain unchanged.

**Limitations:**

Limitations and potential egative impact are adequately discussed.

**Quality:**

3

**Strengths And Weaknesses:**

The paper's principal strength lies in its clear identification of a significant gap in current causal discovery literature—namely, the lack of effective, scalable tools for modeling dynamic causal structures. UnCLe is a relevant contribution: its perturbation-based post-hoc analysis offers a novel approach to quantifying time-varying causal influence, and the architectural design with shared TCN-based encoders/decoders enhances scalability and robustness. The results demonstrate UnCLe’s effectiveness, especially in dynamic scenarios where it outperforms the closest dynamic baseline, GVAR, both qualitatively (e.g., motion phase interpretation) and quantitatively. Overall, the paper is well-written. Although Python code and datasets are provided in the supplementary material, I did not verify the reproducibility of the experiments.

Regarding weaknesses, while the model architecture is novel in its integration of existing components (e.g., TCNs, perturbation-based attribution), it does not introduce fundamentally new neural paradigms. Furthermore, the paper could engage more deeply with theoretical guarantees (e.g., identifiability, spurious effects), and the interpretability of semantic channels. Lastly, the dynamic causal graphs are evaluated largely through visual or intuitive means (e.g., MoCap phases). More rigorous quantitative evaluation on dynamic benchmarks would further strengthen the case.

---

> ### Author Rebuttal · Authors · 2025-07-31
>
> ### **Response to Reviewer T7tD**
>
> We thank you for your positive assessment and for providing valuable feedback to strengthen our paper.
>
> **Weaknesses:**
>
> *   **W1: On the lack of fundamentally new neural paradigms.**
>     We agree that UnCLe does not introduce a fundamentally new neural paradigm. Its novelty lies in the specific architectural integration of established components (like TCNs) and a perturbation-based analysis, tailored specifically to tackle the challenging and under-explored problem of scalable *dynamic* causal discovery.
>
> *   **W2: On the need for more theoretical guarantees and interpretability.**
>     (This point is also raised by Reviewers SUxp and 9Mey). We agree that more theoretical guarantees, such as for identifiability, are an important next step. Please see our detailed response to **Reviewer SUxp, Weakness W1**. Regarding channel interpretability, we provide a detailed analysis in **Appendix I ("Interpreting Channel-wise Causal Contributions")**, which shows how different channels capture different facets of the underlying causal structure.
>
> *   **W3: On the need for more rigorous quantitative evaluation on dynamic benchmarks.**
>     We appreciate this suggestion. In the paper, we provide quantitative evaluation on the dynamic SEM (dSEM) dataset via AUROC/AUPRC and on the MoCap dataset via a missing rate metric for skeletal connections. We also complement this with qualitative analysis of the MoCap phases. We will enhance our qualitative analysis by incorporating more specific biomechanics literature, as suggested by Reviewer K7ML, and will continue to explore additional quantitative evaluation methods for dynamic graphs.
>
> **Questions:**
>
> *   **Q1: On clarifying the link between Dependency Matrices and true causal effects.**
>     We provide a detailed analysis of this in Appendix I ("Interpreting Channel-wise Causal Contributions"), using the Lorenz#1 dataset as an example. In brief, each channel's Dependency Matrix learns a specific "semantic" mode of interaction between variables. The true causal mechanism is modeled by the collective action of all these interaction modes. Aggregating them allows us to recover a comprehensive view of the static causal graph.
>
> *   **Q2: On the sensitivity of UnCLe to the choice of perturbation strategy.**
>     This is a great question. We experimented with three strategies: permutation (our default), zero-masking, and random noise injection. Permutation performed best. We reason that zero-masking disrupts the data distribution, affecting model stability, while noise injection does not fully remove the original signal. Permutation uniquely preserves the marginal distribution while effectively nullifying predictive temporal information. We believe this is a valuable point and will add a summary of this sensitivity analysis to the final version.
>
> *   **Q3: On avoiding spurious correlations in high-dimensional, limited-data settings.**
>     This is a key challenge in causal discovery. UnCLe mitigates this risk through its parameter-sharing TCN autoencoder, which allows the model to learn robust, common representations from the entire dataset, effectively increasing the data available for learning each component. This enhances model stability. We tested this scenario with the Lorenz#3 dataset (N=100, T=500), where UnCLe outperformed all baselines, suggesting our approach is relatively robust in such settings.
>
> *   **Q4: On providing more detailed runtime and memory analysis for scalability.**
>     We agree this is important for assessing practical applicability. UnCLe's scalability stems from its parameter-sharing design and parallelizable computations. In the final version, we will expand the "Time Efficiency" subsection to include a table or plot detailing UnCLe's runtime and memory usage across different data scales (N and T), and supplement the existing Figure 4 with efficiency comparisons on other datasets.
>
> Thank you for your constructive and detailed feedback.

---

> > ### Comment · Reviewer_T7tD · 2025-08-01
> >
> > Thank you for your reply. My score remains unchanged.

---

### Official Review · Reviewer_SUxp · 2025-06-29

**Clarity:** 4
**Significance:** 3
**Originality:** 3
**Rating:** 5
**Confidence:** 4

**Summary:**

UnCLe is a scalable framework for causal discovery in non-linear, high-dimensional time series. A parameter-sharing TCN encoder learns latent features, while a sparse auto-regressive matrix Ψ predicts the next step. Causal graphs are then extracted in two ways:
1. Dynamic – randomly permuting one series and measuring the error increase on others (fast, perturbation-based).
2. Static – aggregating Ψ weights without extra passes.
Across synthetic static/dynamic benchmarks, fMRI, human motion capture, and 300-node traffic data, UnCLe matches or outperforms nine baselines, sits on the AUROC / runtime Pareto frontier, and yields interpretable graphs—yet keeps parameters and runtime modest thanks to weight sharing.

Across synthetic static/dynamic benchmarks, fMRI, human motion capture, and 300-node traffic data, UnCLe matches or outperforms nine baselines, sits on the AUROC / runtime Pareto frontier, and yields interpretable graphs—yet keeps parameters and runtime modest thanks to weight sharing.

**Questions:**

1. How did the framework perform on time series data with missingness or irregularities?
2. Can this post-hoc perturbation method be extended to other similar temporal deep learning frameworks?
3. L1 on Ψ encourages sparsity, but dynamic graphs in nature may densify temporarily; does the regulariser hamper detection of dense episodes?

**Ethical Concerns:**

["NO or VERY MINOR ethics concerns only"]

**Final Justification:**

I thank the authors for their detailed response, which has addressed most of my questions. Considering that the paper is already well-established and the clear and convincing rebuttal, I would like to maintain my rating as 5.

**Limitations:**

The paper’s limitations section is reasonably clear about scalability ceilings and the lack of formal identifiability guarantees, but the broader-impact discussion is essentially missing (the checklist marks “N/A”). I recommend the authors:

1. Add a short societal-impact paragraph: spell out potential misuse risks (e.g., drawing unjustified causal claims in sensitive domains like finance, health, or policing) and emphasize that UnCLe’s graphs should not be used for high-stakes decisions without expert validation.
2. Clarify deployment safeguards: outline recommendations such as thresholding edges by confidence, combining UnCLe with domain knowledge, or performing prospective validation before operational use.

Incorporating these points would more fully address the negative societal impact while rewarding the authors’ transparency.

**Paper Formatting Concerns:**

Non major formatting issues.

**Quality:**

3

**Strengths And Weaknesses:**

Strengths:
1. Novel deep learning method explicitly targeting time-resolved causal graphs in large, non-linear systems.
2. Perturbation yields accurate dynamic edges; aggregation gives an immediate static summary without extra passes—useful in latency-constrained settings.
3. Extensive evaluation on synthetic (static and dynamic), medium-dimensional medical data, high-dimensional traffic & biomechanics; baselines span classical, score-based, and neural Granger families.

Weaknesses:
1. No identifiability guarantee: the method relies on predictive perturbations but does not analyse conditions (e.g., faithfulness, invertibility) under which the framework recovers the true causal strength.
2. Ψ operates only on lag 1. Although the authors use dilated TCNs, empirical degradation when true lags > 1 is not studied.
3. Error gains depend on permutation; robustness to alternative perturbations (masking, noise injection) is unclear.

---

> ### Author Rebuttal · Authors · 2025-07-31
>
> ### **Response to Reviewer SUxp**
>
> We sincerely thank you for your detailed review and for highlighting the strengths of our work. We appreciate the opportunity to clarify the points you raised.
>
> **Weaknesses:**
>
> *   **W1: On the lack of identifiability guarantees.**
>     We concur that formal identifiability guarantees are crucial for causal discovery methods. This is a well-known and challenging open problem for many deep learning-based approaches, whose theoretical properties often lag behind their empirical success. While designing UnCLe, we explored connections to causal disentanglement literature (e.g., [1][2]), but found that proving formal properties for our specific architecture was non-trivial. Therefore, this work focuses on providing strong and extensive empirical validation of UnCLe's effectiveness. We have added providing a formal identifiability analysis as a primary direction for future work.
>
> *   **W2: On not studying cases where lag > 1.**
>     (This point is also raised by Reviewer 9Mey). Please see our detailed response to **Reviewer 9Mey, Weakness W1**, which includes experimental results for different lags. In summary, the TCN architecture's receptive field allows the model to handle long-range dependencies, and we found that explicitly increasing the autoregressive lag in the Dependency Matrices did not improve, and sometimes hindered, performance, likely due to parameter redundancy.
>
> *   **W3: On the robustness of permutation compared to other perturbation methods.**
>     (This point is also raised by Reviewer T7tD). Please see our detailed response to **Reviewer T7tD, Question Q2**. In summary, we experimented with permutation, zero-masking, and random noise injection. Permutation yielded the best results because it effectively removes temporal information while preserving the variable's marginal distribution, ensuring model stability. We will add a summary of this sensitivity analysis to the final paper.
>
> *   **W4: On the performance drop on sparse synthetic datasets with >50% missing edges.**
>     We appreciate you pointing this out for clarification. However, we were unable to locate the specific statement about ">50% missing edges" in our manuscript. Could you please help clarify which part of the paper this refers to? We would be very grateful for the opportunity to address this point more directly.
>
> **Questions:**
>
> *   **Q1: On handling missing or irregular data.**
>     This is an important practical consideration. Our current work assumes complete and regularly-sampled data. For datasets with missingness or irregularities, UnCLe would require a standard preprocessing step, such as imputation, before application. Explicitly handling such data within the model architecture is a valuable direction for future research.
>
> *   **Q2: On extending the post-hoc perturbation method to other frameworks.**
>     This is a very keen observation. Yes, the post-hoc perturbation framework is in principle model-agnostic and could be applied to other multivariate time-series forecasting models. Exploring its effectiveness across different architectures is a promising research avenue that we hope our work will inspire.
>
> *   **Q3: On whether L1 regularization hampers the detection of temporarily dense graphs.**
>     This addresses a fundamental trade-off between interpretability and complexity. Our goal with L1 regularization is to encourage the model to capture the most stable and dominant causal relationships. While a very strong regularizer could potentially suppress transient dense connections, we found that a moderate L1, combined with the aggregation of diverse patterns across multiple channels, allows the model to represent complex structures without being overwhelmed by noise or spurious connections. The key is to find a balance that yields an interpretable yet sufficiently descriptive graph.
>
> **Limitations:**
>
> *   **On adding a discussion of broader impacts and safeguards.**
>     Thank you for this crucial feedback. We agree that a discussion on broader impacts is essential. In the final version of this paper, we will add a dedicated section to address the potential misuse of causal claims in sensitive domains and outline best practices for responsible application, such as combining UnCLe with domain expert knowledge and performing prospective validation.
>
> Thank you again for your thoughtful review.
>
> ----
>
> **References**
>
> \[1] Song, Xiangchen, et al. "Temporally disentangled representation learning under unknown nonstationarity.", NeurIPS 2023
>
> \[2] Song, Xiangchen, et al. "Causal temporal representation learning with nonstationary sparse transition.", NeurIPS 2024

---

> > ### Comment · Reviewer_SUxp · 2025-08-01
> >
> > Thank you very much for the detailed response. You can simply ignore weakness 4; sorry for any confusion caused.
> > My rating remains unchanged.

---

### Official Review · Reviewer_K7ML · 2025-07-01

**Clarity:** 4
**Significance:** 3
**Originality:** 3
**Rating:** 5
**Confidence:** 4

**Summary:**

The article proposes UnCLe, a method for dynamic causal discovery. The method using an auto-encoding type structure to learn a latent representation of a nonlinear time series which linearizes the underlying dynamics. The method is evaluated on several interesting synthetic and real-world examples. Experimental results demonstrate the method effectively and suggest it to be an important contribution to the field.

**Questions:**

- Why are the static and dynamic graphs learned by apparently two distinct mechanisms? Can the authors comment on why they proposed two distinct methods in this article?

- How do we verify linearity in latent dynamics is a good assumption? Linear dynamics are NOT sufficient to describe nonlinear phenomena. That said, linear models can be useful to represent nonlinear dynamics when the correct representation is used (Brunton et al, 2017). In that sense, UnCLe could argued for quite nicely using existing theory. I recommend that the authors take a look at some of the literature on the linearization of nonlinear dynamic systems (e.g., (Socha, 2007)) to understand the limitations of their proposed method, and to consider these ideas in future work.

- It should be emphasized in the article that the reason that the UnCLe(P) method can detect causation from prediction is specifically because it models the causal mechanism that generates the data. Hence, if the predictions are inaccurate, it indicates that the assumed causal dynamics did not anticipate the next state, which is one consequence of the graph changing over time.

### References

Brunton, Steven L., et al. "Chaos as an intermittently forced linear system." Nature communications 8.1 (2017): 19.

Socha, Leslaw. Linearization methods for stochastic dynamic systems. Vol. 730. Springer, 2007.

**Ethical Concerns:**

["NO or VERY MINOR ethics concerns only"]

**Final Justification:**

i am satisfied with the authors responses.

**Limitations:**

I believe the authors commented sufficiently on the limitations of the work, apart from my requests above.

**Quality:**

4

**Strengths And Weaknesses:**

The article is clearly written and easy to follow. The proposed method is interesting and well-detailed. The authors provide compelling motivation, and the experiments clearly support the proposed method.

I only identified minor weaknesses in the article, which I have detailed below.
- UnCLe(P) appears to be uniformly better than UnCLe(A). The authors state that UnCLe(A) might be preferable when speed is desired for causal discovery. But when would I want speed? Causal discovery is often a scientific task, performed offline on pre-recorded data sets, and it's hard for me to imagine when I would need to speed up this procedure. Also, it's not clear that the gain in time that I get from using UnCLe(A) vs UnCLe(P) is significant relative to the overhead of training the Uncoupler\Recoupler networks. Figure 4 suggests that running UnCLe already takes a few minutes. Does this speed up significantly if I use (A)? Please comment on this.
- The experiment with the dynamic SEM would benefit from a figure that compares the "ground truth" causal strength to the UnCLe causal strengths, and to comment on whether they align or not, and why the result is\isn't reasonable.
- Which version of UnCLe is used in Figure 4?
- The discussion surrounding the MoCap experiment would benefit from adding references to existing literature that explains how the UnCLe method produced a "better" time-varying graph. Without a ground truth, asserting that the UnCLe graph is preferrable requires the reader to have some kind of knowledge of MoCap. In the worst case, it might be true that dense graphs are to be expected in MoCap and that the sparsity of the UnCLe graphs is undesirable. Please comment on this.

---

> ### Author Rebuttal · Authors · 2025-07-31
>
> ### **Response to Reviewer K7ML**
>
> We sincerely thank you for your insightful review and positive evaluation of our work. We appreciate the constructive feedback and will address the identified points to improve the paper.
>
> **Weaknesses:**
>
> *   **W1: On the time sensitivity of causal discovery and the speed-up from weight aggregation (UnCLe(A)).**
>     That is an excellent point. While many scientific discovery tasks are performed offline, latency can be critical in applications like real-time monitoring in industrial systems. The speed-up from UnCLe(A) is most significant on large-scale datasets where the perturbation process (UnCLe(P)) becomes more time-consuming. For smaller datasets, the overhead of perturbation is negligible. In the final version, we will add an analysis comparing the post-hoc inference times for UnCLe(P) and UnCLe(A) across different data scales to better quantify this trade-off.
>
> *   **W2: On visualizing the ground truth (GT) causal strength in the DSEM experiment.**
>     We agree that a direct comparison would be ideal for interpretability. The causal *direction* is explicitly defined in the SEM equations, and we visualized this with color-coded segments and vertical lines. However, we did not plot a GT "causal strength" line because the SEM coefficients, while strongly correlated with strength, are not directly equivalent to a singular strength value, making a direct GT plot potentially ambiguous. Our visualization approach, which includes smoothing the discovered strengths, aims to clearly show how UnCLe correctly identifies the switching dominance between variables in alignment with the ground truth mechanism.
>
> *   **W3: On clarifying which version of UnCLe was used in Figure 4.**
>     Our apologies for the omission. Figure 4 shows the results for the perturbation-based method, **UnCLe(P)**. We will clarify this in the caption in the final version.
>
> *   **W4: On substantiating the MoCap analysis with domain literature.**
>     We thank the reviewer for this insightful and critical question. The reviewer is entirely correct that without a ground truth, asserting a causal graph is "better" is a non-trivial claim, and we agree our argument must be substantiated with external domain knowledge.
>     We will revise the manuscript to incorporate this justification. Our core argument, supported by biomechanics literature (e.g., Hara et al., 2008; DeVita & Skelly, 1992), will be that UnCLe's graphs are "better" because their "structural evolution" aligns with the known, "phase-specific coordination strategies" of the jump. Specifically, UnCLe captures the shift from full-body coordination (in crouch/landing) to localized adjustments (in flight), a dynamic nuance missed by the baselines. We will add the necessary analysis and citations in the final version to fully support this interpretation.
>
> **Questions:**
>
> *   **Q1: On the motivation for proposing two distinct causal discovery mechanisms (static/dynamic).**
>     This is a great question about our design process. Our work began with developing a robust representation and prediction model for time series. We first explored perturbation for static discovery. In seeking to explain its success, we found that the learned Dependency Matrices themselves contained a static causal blueprint (UnCLe(A)). We then realized the perturbation approach naturally extended to dynamic discovery (UnCLe(P)).
>     We chose to present both because they are complementary. UnCLe(P) is our primary contribution for accurate dynamic discovery. UnCLe(A) serves as a zero-cost, interpretable diagnostic tool derived directly from the model's parameters, providing a valuable static summary and a basis for understanding the model's learned structure. We believe both approaches offer valuable insights to the community.
>
> *   **Q2: On validating the assumption of linear latent dynamics.**
>     Thank you for pointing us to this highly relevant literature. We agree that framing our work within the context of dynamical systems theory, such as linearization of nonlinear systems, would greatly strengthen the paper's theoretical underpinnings. This is a crucial point for understanding the limitations of our method and for guiding future work. We will incorporate this perspective and the suggested references into our discussion.
>
> *   **Q3: On emphasizing the mechanism behind perturbation-based discovery.**
>     This is a key insight. We will revise the methodology section to more explicitly state that UnCLe(P)'s effectiveness stems from its ability to model the data's causal generative mechanism. An inaccurate prediction under perturbation indicates that the model's learned causal dynamics did not anticipate the next state, which is a direct consequence of disrupting a true causal link.
>
> Thank you again for your valuable time and constructive suggestions.

---

### Comment · Area_Chair_6EJn · 2025-08-01
**The time to start author-reviewer discussions**

Dear all reviewers,

The author rebuttal period has now concluded, and authors' responses are
available for the papers you are reviewing. The Author-Reviewer Discussion
Period has started, and runs until August 6th AoE.

Your active participation during this phase is crucial for a fair and
comprehensive evaluation. Please take the time to:

- Carefully read the author responses and all other reviews.
- Engage in a constructive dialogue with the authors, clarifying points,
  addressing misunderstandings, and discussing any points of disagreement.
- Prioritize responses to questions specifically addressed to you by the authors.
- Post your initial responses as early as possible within this window to
  allow for meaningful back-and-forth discussion.

Your insights during this discussion phase are invaluable.
Thank you for your continued commitment to the NeurIPS review process.

Bests,
Your AC

---

### Note · Authors · 2025-08-13

We sincerely thank all reviewers and the Area Chair for their time and exceptionally constructive feedback. The detailed discussion has been invaluable for refining our work and clarifying our contributions.

In our responses, we have aimed to provide comprehensive clarifications on the points raised. Specifically:

- **We have provided detailed explanations** on our methodological choices, including the complementary roles of UnCLe(P) and UnCLe(A), the trade-offs of L1 regularization, the learning strategy for the Dependency Matrices ($\boldsymbol{\Psi}$), its parameterization, and the novelty of our architectural integration. We also addressed the model's robustness regarding spurious correlations, its application to sparse datasets, and the interpretation of the DSEM visualizations.

- **We have also engaged in an open discussion** on the core theoretical underpinnings of our work. We acknowledged that formal identifiability remains a challenging open problem for deep learning methods, and we contextualized our linear latent space assumption with the suggested literature on dynamical systems, framing these as key directions for future research.


Building on this productive discussion, we are committed to incorporating the following key revisions into the final version of the paper to enhance its rigor, clarity, and completeness:

1. Textual and Framing Enhancements: We will strengthen the explanation of the causal discovery mechanism in the methodology section and add a dedicated **"Broader Impacts" section** to the main paper.

2. New Experimental Analyses in the Main Paper: We will add new ablation studies to investigate the effects of **higher-order lags** and the model's sensitivity to different **perturbation strategies**. We will also expand the "Time Efficiency" section with a more detailed **runtime and scalability analysis**.

3. Strengthened Evaluation and Appendix: We will substantiate our MoCap analysis by integrating **citations from biomechanics literature**. In the appendix, we will add a **time-cost comparison** for the post-hoc inference of UnCLe(P) vs. UnCLe(A).


We are confident that the combination of the clarifications provided and these planned revisions will substantially strengthen the paper. We are grateful for the opportunity to improve our work and thank you once more for your valuable guidance.

---

### Decision · Program_Chairs · 2025-09-17

**Decision:**

Accept (poster)

**Comment:**

This paper introduces UnCLe, a novel and scalable deep learning framework
for dynamic causal discovery in non-linear temporal systems. The method
uses an "Uncoupler-Recoupler" architecture to learn disentangled latent
representations and infers time-varying causal graphs by analyzing
prediction errors from a post-hoc perturbation analysis.

The reviewers were in strong agreement on several key strengths of the
work. They praised the paper for addressing a significant and
under-explored problem, noting that the ability to model dynamic, rather
than static, causal relationships is a critical step forward for
understanding complex real-world systems. The reviewers found the proposed
method to be novel, well-motivated, and clearly presented. A major point of
consensus was the extensive and compelling empirical evaluation across a
diverse set of synthetic and real-world benchmarks (including fMRI and
human motion capture), where UnCLe consistently outperforms
state-of-the-art baselines.

The primary concern raised during the review process was the lack of formal
identifiability guarantees—a well-known and challenging open problem for
many deep learning-based causal discovery methods. The authors forthrightly
acknowledged this limitation. Other concerns regarding sensitivity to
hyperparameters and model design choices were effectively addressed in the
author rebuttal, often with new ablation studies.

Overall, despite the theoretical limitations, the paper's significant
contribution to the important problem of dynamic causality, backed by
strong and comprehensive empirical results, makes a compelling case for
acceptance. UnCLe represents a significant practical advance and is likely
to be of high interest to the community.